# ∇QDARTS: Quantization as an Elastic Dimension to Differentiable NAS

**Payman Behnam*[†], Uday Kamal*, Sanjana Vijay Ganesh, Zhaoyi Li, Michael Andrew Jurado, Alind Khare**
*Georgia Institute of Technology*

**Igor Fedorov**
*Meta*

**Gaowen Liu**
*Cisco Research*

**Alexey Tumanov**
*Georgia Institute of Technology*

**Reviewed on OpenReview:** *https://openreview.net/forum?id=ubrOSWyTS8*

## Abstract

Differentiable Neural Architecture Search methods efficiently find high-accuracy architectures using gradient-based optimization in a continuous domain, saving computational resources. Mixed-precision search helps optimize precision within a fixed architecture. However, applying it to a NAS-generated network does not assure optimal performance as the optimized quantized architecture may not emerge from a standalone NAS method. In light of these considerations, this paper introduces ∇QDARTS, a novel approach that combines differentiable NAS with mixed-precision search for both weight and activation. ∇QDARTS aims to identify the optimal mixed-precision neural architecture capable of achieving remarkable accuracy while operating with minimal computational requirements in a single-shot, end-to-end differentiable framework, obviating the need for pretraining and proxy methods. Compared to fp32, ∇QDARTS shows impressive performance on CIFAR10 with (2,4) bit precision, reducing bit operations by 160× with a slight 1.57% accuracy drop. Increasing the capacity enables ∇QDARTS to match fp32 accuracy while reducing bit operations by 18×. For the ImageNet dataset, with just (2,4) bit precision, ∇QDARTS outperforms state-of-the-art methods such as APQ, SPOS, OQA, and MNAS by 2.3%, 2.9%, 0.3%, and 2.7% in terms of accuracy. By incorporating (2,4,8) bit precision, ∇QDARTS further minimizes the accuracy drop to 1% compared to fp32, alongside a substantial reduction of 17× in required bit operations and 2.6× in memory footprint. In terms of bit-operation (memory footprint), ∇QDARTS excels over APQ, SPOS, OQA, and MNAS with similar accuracy by 2.3× (12×), 2.4× (3×), 13% (6.2×), 3.4× (37%), for bit-operation (memory footprint), respectively. ∇QDARTS enhances the overall search and training efficiency, achieving a 3.1× and 1.54× improvement over APQ and OQA, respectively.

## 1 Introduction

Neural Architecture Search (NAS) has emerged as a powerful method that reduces the manual and human effort to find optimum network architectures for the target applications. However, traditional NAS methods require complete training of every candidate network, which is computationally prohibitive due to the enormous search space (Kyriakides & Margaritis (2020)). To alleviate this problem, several bodies of work

---

*[*]Equal contribution.*
*[†]Corresponding author(payman.behnam@gatech.edu).*

suggest different techniques to reduce the NAS search space (Negrinho & Gordon (2017); Liu et al. (2018a; 2017a); Kandasamy et al. (2018)). Evolutionary search, Monte-carlo tree search (MCTS) (Negrinho & Gordon (2017)), sequential model-based optimization (SMBO) (Liu et al. (2018a; 2017a)), and Bayesian optimization (Kandasamy et al. (2018)) are examples of these approaches. However, even these approaches suffer from inherent search inefficiency, because they treat architecture search as a black-box optimization problem over a *discrete domain*, which leads to a large number of architecture evaluations. These methods are also criticized for their high time complexity, energy consumption, and not being scalable (Elsken et al. (2019); Liu et al. (2018b)).

To address these challenges, weight-sharing methods have been proposed (Ullrich et al. (2017); Cai et al. (2019); Sahni et al. (2021); Liu et al. (2018b); Xu et al. (2019)). These methods provide the advantage of sharing weights across varied architectures, resulting in a considerable reduction in the compute and memory resources needed to conduct the search. As a result, they have recently attracted significant attention ( Ullrich et al. (2017); Cai et al. (2019); Sahni et al. (2021); Liu et al. (2018b); Xu et al. (2019); Sahni et al. (2021); Yu et al. (2020a;a;b); Banbury et al. (2021)).

Among the weight-sharing methods, DARTS (Liu et al. (2018b)) addresses the challenge of scalability of the architecture search through a differentiable search formulation. It relies on *continuous relaxation* of the architectural representation by employing efficient search using gradient descent compared with conventional approaches that use evolution, reinforcement learning, sampling, and random searches over discrete and non-differentiable search spaces.

DARTS has introduced a novel avenue of research in the NAS domain by circumventing the need to traverse discrete and large search spaces ( Xu et al. (2019); Wan et al. (2020); Zela et al. (2019); Chu et al. (2020); Liang et al. (2019); Chen & Hsieh (2020); Wang et al. (2021); Zhang et al. (2021); Yang et al. (2021); Zhou et al. (2020); Liu et al. (2021); Zhang et al. (2020a); Chen et al. (2019)). Among DARTS methods, PC-DARTS (Xu et al. (2019)) further reduces the search cost by introducing partial channel sampling during the search operation. As a result of this strategic implementation, the framework has improved memory efficiency and accelerates processing speeds (Elsken et al. (2019)).

Although existing NAS methods can find a suitable architecture for a target device and application, they suffer from computationally expensive floating point operations. These operations are cost-prohibitive, especially for resource-constrained devices. Existing research work demonstrates that lower-bit quantization policy improves the performance of the network in terms of memory footprint, latency, computational cost, bandwidth efficiency, power, and energy consumption ( Krishnamoorthi (2018); Choukroun et al. (2019); Wang et al. (2019); Gholami et al. (2021)). However, applying quantization is not straightforward in a NAS setting. First, due to the search space explosion, it is non-trivial to incorporate quantization during architecture/weight search (Wang et al. (2019; 2020); Bai et al. (2021)).

Few methods have been proposed to incorporate quantization into NAS. These methods usually exploit proxies (e.g., pretrained neural networks) to estimate the performance of the sampled architecture/quantization search (Wang et al. (2020)). Proxy-based methods usually lack accurate estimation. In addition, proxies need to be treated carefully to ensure that quantized architectures are ranked fairly. Besides, evaluation using a proxy is a time-consuming procedure (Bai et al. (2021)).

Second, it is difficult to predict, in advance, exactly how quantization will affect a neural network found by a NAS method. The reason is that quantization efficiency depends on the neural architecture itself. For instance, Shen et al. show that the accuracy of MobileNetv2 with full precision (fp32) is better than ResNet18. However, with the 2-bit precision, the ResNet18 accuracy is higher than MobileNetv2 (Shen et al. (2021)).

Third, it is unclear how quantization should be applied within NAS architectures. As an example, we can apply quantization to a regiment of potential *SubNets* within a DARTS network that results in a multitude of quantized architectures. However, there is no guarantee that the resultant *SuperNet*, which is formed by combining these *SubNets*, will have the optimal quantized architecture.

In this paper, we introduce $\nabla$`QDARTS`, **a novel framework that integrates architecture and weight optimization along with a comprehensive mixed-precision search for both weights and activations**. The framework is achieved in a single shot, end-to-end, differentiable optimization, eliminating the need

for any pretraining or proxy. Furthermore, $\nabla$QDARTS leverage the complexity-aware training to favor the discovery of lower computational complexity (i.e., lower-bit precision) models during the search phase. In this manner, the model produced by the search stage is optimized for a computational budget determined by a user-specified hyperparameter where no grid search is required. In addition, $\nabla$QDARTS enables a trade-off between bit-operation (*BitOps*) and accuracy to make it a suitable fit for the target edge application with relatively low GPU hours.

In summary, the contributions of the paper are as follows:

- We propose $\nabla$QDARTS, which provides a single-shot, end-to-end, differentiable framework by integrating architecture and weight search with a mixed-precision quantization policy for both weights and activations, without any need for pretraining and proxies.

- $\nabla$QDARTS provides a framework for flexible navigation in accuracy-BitOps trade-off space, where the produced model by the search stage is optimized for a complexity budget determined by a user without any grid-search requirement.

- $\nabla$QDARTS enables highly efficient and accurate architecture discovery even when ultra low bit-precision is considered (2 and 4) with negligible accuracy loss compared to the full-precision (fp32) networks.

## 2 Background and Challenges

### 2.1 Differentiable NAS Formulation

The key objective of any NAS framework is to find both the optimal architecture, $\alpha^*$, and its corresponding weights, $\omega^*$, which is a bi-level optimization problem formulated by the following equations:

$$\min_{\boldsymbol{\alpha}} \mathcal{L}_{\text{val}}(\omega^*(\alpha), \alpha), \text{s.t.} \quad \omega^*(\alpha) = \operatorname*{argmin}_{\boldsymbol{\omega}} \mathcal{L}_{\text{train}}(\omega, \alpha) \tag{1}$$

where $\mathcal{L}_{val}(.,.)$ and $\mathcal{L}_{train}(.,.)$ denote the training and validation loss, respectively. The architecture parameters are optimized over the validation dataset to avoid overfitting (Liu et al. (2018b)). To ensure differentiability, discrete architecture search spaces are converted to continuous ones (Wu et al. (2019a); Zhang et al. (2020b); Zela et al. (2019); Cai et al. (2018)). In continuous relaxation, the architecture search space $A$ is relaxed to $A(\theta)$, where $\theta$ is the continuous parameter that presents the distribution of architectures ($A \subseteq A(\theta)$). In this way, we can employ gradient-based methods to find the best weight and architecture (Guo et al. (2020)).

To solve these equations, we can use either single-level or bi-level optimizations. In single-level optimization, $\alpha$ and $\omega$ are updated simultaneously in each step. While this is computationally less expensive, they suffer from poor generalization due to overfitting. Bi-level optimization, on the other hand, updates $\alpha$ and $\omega$ in an alternative fashion.

### 2.2 Quantization

The quantization can be fixed-bit, meaning the entire network has a single weight and/or activation precision. Because different layers, filters, and channels have various impacts on the accuracy, a model's accuracy can be significantly degraded when it is uniformly quantized to ultra-low precision. This problem can be addressed with mixed-precision quantization schemes (Dong et al. (2019); Habi et al. (2020); Hu et al. (2021); Zhao et al. (2021)). In mixed-precision, each layer, filter, channel, and/or activation have a separate precision. It has been shown that converting floating-point values to low-precision fixed integer values in four bits or less can reduce memory footprint and latency significantly (Gholami et al. (2021); Dong et al. (2019); Deng et al. (2020)). For instance, ResNet50 inference with INT4 precision can produce a speedup of 50%-60% compared to INT8 inference, emphasizing the importance of using lower-bit precision for optimal performance (Salvator et al. (2019)). However, when the bit precision goes below 8-bit, reaching high accuracy, especially for large datasets, is a challenging task (Sun et al. (2020; 2022)).

Table 1: Comparing ∇QDARTS with state-of-the-art quantization methods. "Weight-sharing" means whether the method is utilizing a weight-sharing solution. "Single-shot mixed-precision" quantization policy refers to using QAT to train the *SuperNet* with arbitrary mixed-precision quantization policies. "Mixed-precision" means that the method supports different precisions. "No training during search" means during the search phase there is no need for re-training the sampled network candidate. As ∇QDARTS does not need any sampling mechanism, we do not need to train and evaluate the selected *SubNet*. The research community has set a standard for evaluating NAS methods, using "DARTS Search Space" that has been widely used. It has larger search space compared to the MobileNet search space (Mehta et al. (2022); Zhang & Ding (2023)). "Proxies Avoidance" refers to the fact that the employed technique avoids employing a trained neural network to estimate the accuracy of a sampled architecture. "Pretraining Avoidance" means the method can avoid any required pretraining.

| Property | DNAS 2018 | HAQ 2019 | EdMIPS 2020 | APQ 2020 | OQA 2021 | BatchQuant 2021 | SPOS 2020 | AutoNBA 2021 | MNAS 2019 | UDC 2022 | ∇QDARTS 2025 |
|---|---|---|---|---|---|---|---|---|---|---|---|
| Joint DNN & Quan Search | | | | ✔ | ✔ | ✔ | | ✔ | ✔ | ✔ | ✔ |
| Weight Sharing | | | | ✔ | ✔ | ✔ | ✔ | ✔ | ✔ | ✔ | ✔ |
| Mixed Precision We. Quan | ✔ | ✔ | ✔ | ✔ | | ✔ | ✔ | | ✔ | ✔ | ✔ |
| Mixed Precision Act. Quan | ✔ | ✔ | ✔ | ✔ | | ✔ | ✔ | | ✔ | | ✔ |
| Single-shot Mixed Precision | | | | | | ✔ | ✔ | | | | ✔ |
| Differentiable Quan | ✔ | ✔ | ✔ | | | | | | | ✔ | ✔ |
| DARTS Search Space | | | | | | | | | | | ✔ |
| No Training During Search | | | | ✔ | ✔ | ✔ | ✔ | | ✔ | ✔ | ✔ |
| Proxies Avoidance | | | | | ✔ | ✔ | | | | ✔ | ✔ |
| Pretraining Avoidance | | | | | ✔ | | | | | ✔ | ✔ |
| Relatively Low Train Time | ✔ | ✔ | ✔ | ✔ | ✔ | | | | ✔ | ✔ | ✔ |

## 2.3 Incorporating Quantization into NAS

Few studies try to incorporate quantization into NAS or employ the idea of the differentiable search for efficient quantization of a fixed architecture.

DNAS (Wu et al. (2018)) proposes a differentiable mixed-precision solution per layer that takes a fixed architecture, like ResNet, and searches over the possible quantization bit precision per layer. The method assigns a probability weight to the quantization 'path' per layer such that the entire network is fully differentiable. DNAS is not like a NAS architecture, rather, it employs NAS formula for a fixed architecture to find a proper mixed-precision solution.

HAQ (Wang et al. (2019)) proposes a Hessian-aware mixed-precision quantization technique that finds the optimum bit-precision for each output channel within every layer. The importance and bit precision of each output channel are recognized based on the Hessian values of the channel. To make this method work, it is necessary to have an already trained, fixed architecture network.

EdMIPS (Cai & Vasconcelos (2020)) opts for mixed-precision search strategies for the different filters in a fixed architecture and achieves high efficiency by sharing weights for all mixed-precision operations. It employs the idea of DARTS for finding the best precision of a predefined architecture. EdMIPS employs a block that is a weighted sum of all quantized weights for a single convolution. With $N$ different precisions, this reduces the complexity of the forward and backward passes by nearly a factor of $N$. Similar to DNAS, it does not perform an architecture search on the layer types and just optimizes mixed-precision for a fixed architecture.

APQ (Wang et al. (2020)) applies a joint search for NAS, pruning, and quantization policies. This technique enables mixed-precision architecture search but relies on a proxy to decide the best precision and pruning scheme. It employs an already trained 32-bit precision predictor as a proxy. For training the accuracy predictor, APQ performs QAT on 5000 architectures and quantization policies for 0.2 GPU hours each. Producing this predictor is expensive in terms of time and resource consumption.

OQA (Shen et al. (2021)) develops Once Quantized for All, which achieves training efficiency by gradually shrinking the bit-width of the supernetwork after it trains via bit-inheritance to enable efficient searching for different quantization bit widths. However, this approach lacks the support of mixed-precision search.

BatchQuant (Bai et al. (2021)) proposes a mixed quantization for OFA-like architecture ( and not differentiable architectures). However, it needs pretraining the supernet at the beginning, otherwise, the accuracy will drop. BatchQuant first trains *SuperNet* with bit width options 2, 3, 4, 32 for the first 65 epochs and then continues training the *SuperNet* with 2, 3, 4 bit width options for the rest of the training to reach high accuracy. Besides, due to retraining, the total GPU hours are significantly high. Unlike $\nabla$**QDARTS**, it was proposed for the MobileNet search space.

SPOS (Guo et al. (2020)) employs uniform path sampling for supernet training, conducting random sampling of block and weight bit widths. The evolutionary step determines the final values. However, under SPOS, only channel searches are permitted, reducing the architecture search space size.SPOS targets fixed architecture like ResNet18 and ResNet34.

UDC (Fedorov et al. (2022)) does not support activation quantization. In addition, unlike $\nabla$**QDARTS** that enables lower bit quantization, UDC utilizes higher bit precision like 8- and 32-bit for the weight quantization.

MNAS (Gong et al. (2019)) combines NAS and quantization over the MobileNet search space. MNAS relies on an 8-bit pretrained model and cannot reach high accuracy.

Except for the Batchquant, OQA, and UDC, all of the baselines rely on a proxy estimator. The proxy is usually a trained neural network that predicts the accuracy of the selected architecture. Using proxy increases search space exploration time.

Table 1 shows various features of different techniques. It illustrates how $\nabla$**QDARTS** is unique compared to the other existing methods. Here, different components of the proposed solutions are explained. As seen, $\nabla$**QDARTS** **provides a single-shot, end-to-end differentiable NAS joint with mixed-precision quantization search for both weight and activation without any need for pretraining or proxy**.

## 3 Proposed $\nabla$**QDARTS**

### 3.1 $\nabla$QDARTS **Overview**

We propose $\nabla$**QDARTS**, a differentiable NAS framework that jointly finds optimal architecture, weight, and bit-precision of both weight and activation in a single-shot, end-to-end manner. $\nabla$**QDARTS** extends the existing DARTS (Liu et al. (2018b); Xu et al. (2019)) framework by adding differentiable precision search. Unlike many existing multi-stage methods where architecture search and optimal quantization policy search are explored independently, joint optimization in $\nabla$**QDARTS** allows for exploring more efficient architecture while preserving its accuracy even in a lower precision regime.

Given a target task, we first follow the fixed architectural macroblocks (or cells) similar to (Liu et al. (2018b)) where each cell has a set of architectural components (i.e., convolution with different kernels, pooling, etc.) $o(.) \in \mathcal{O}$. The output of each cell, $c_o$, is a weighted summation of all the candidate operations:

$$c_o = \sum_{o \in \mathcal{O}} \alpha'_o o(x) \tag{2}$$

where $\alpha'_o$ is a set of learnable weights defined as:

$$\alpha'_o = \frac{\exp\left(\alpha_o\right)}{\sum_{o_i \in \mathcal{O}} \exp\left(\alpha_{o_i}\right)} \tag{3}$$

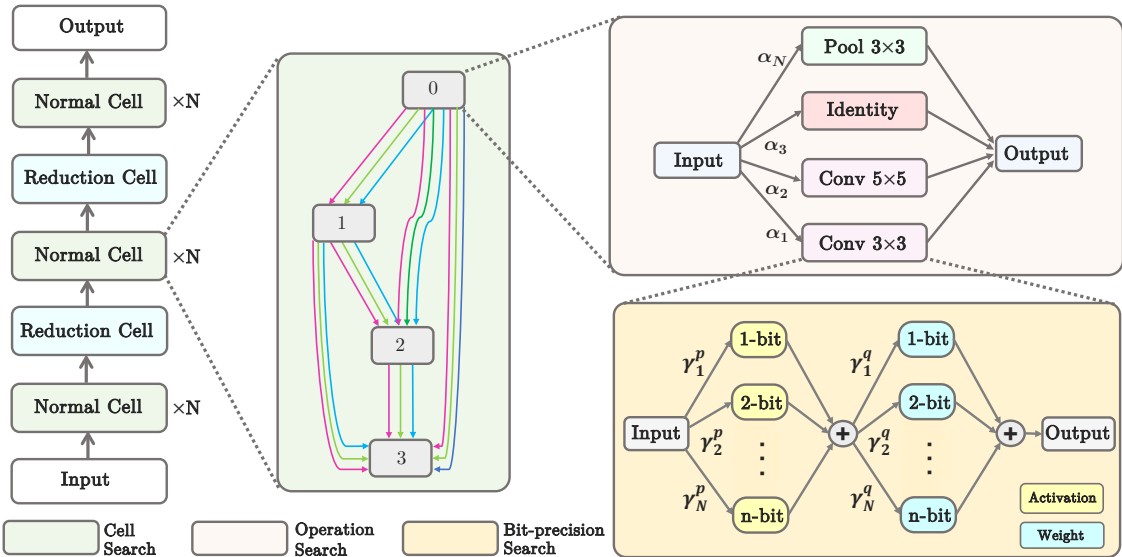

Figure 1: ∇QDARTS brings quantization into DARTS. Each cell consists of the operation search of DARTS, while the innermost block shows the mixed precision search candidates. Crucially, the entire *SuperNet* is fully differentiable. Hence, the architecture and mixed precision search can be performed simultaneously in an end-to-end single-shot fashion.

Thus the architecture search involves learning this set of continuous variables that define the relative weights of each operation over the same input. Next, consider a specific operation with $w$ weight and $a$ activation with $n_a$ and $n_w$ possible bit precisions for activation and weight, respectively. The output of this operation can be formulated as:

$$y = \sum_{i=1}^{n_a} \gamma_i^p a_i \left( \sum_{j=1}^{n_w} \gamma_j^q w_j(x) \right) \tag{4}$$

$$\text{s.t. } \sum \gamma_i^p = 1, \sum \gamma_j^q = 1, \gamma^p, \gamma^q \in [0,1]$$

Here $\gamma_i^p$ and $\gamma_j^q$ are the weighting hyperparameters for mixed-precision operations (with $\{p, q\}$ as the set of learnable parameters) for the $i$-th and $j$-th precision path of activation and weight, respectively, computed through the following equations:

$$\gamma_i^p = \frac{\exp(p_i)}{\sum_k \exp(p_k)}, \quad \gamma_j^q = \frac{\exp(q_j)}{\sum_k \exp(q_k)} \tag{5}$$

Similar to the architectural search, we leverage continuous relaxation of the bit-precision search space. While the most straightforward way is to combine search over architecture and quantization bit precision space in a multi-stage fashion (i.e., find the optimal architecture in full precision, then search over the bit-precisions), we have experimentally shown in Section 4 how such naive integration results in suboptimal performance. The key challenge in incorporating such mixed-precision search into the NAS framework is that simultaneous exploration of different quantization bit-precisions per candidate operations can make the search space computationally intractable. To address this challenge, we extend the originally proposed bi-level optimization in DARTS (Liu et al. (2018b)) to combine both weight and bit-precision as a unified lower variable (details in the following Subsection). Furthermore, to constrain the search space, we leverage the channel-masking mechanism proposed in PC-DARTS (Xu et al. (2019)) to reduce the search cost by 1/K, where K is the number of channels for each convolution operation ($K >> 1$).

Another challenge to search over both architecture and precision parameters is that optimizing them over the same objective function (i.e., the validation loss) would always favor higher precision. Therefore, to penalize the optimization for high compute cost, we additionally add model complexity as a regularizing term to the

loss function to incentivize the discovery of model architectures with lower computational complexity during the search phase.

Figure 1 shows the overall perspective of the $\nabla$`QDARTS`. The mixed-precision quantization in $\nabla$`QDARTS` considers all normal and reduction cells in DARTS search space, where different normal cells and/or reduction cells can have different bit precision for both weight and activation.

### 3.2 $\nabla$`QDARTS` Optimizations

Our formulation ideally involves optimization over three levels of variables: architecture parameters ($\alpha$), weights ($w$), and bit-precision ($\gamma$). While it can be desired to add $\gamma$ as another level of inner variable to make it a tri-level optimization problem, this would increase the optimization steps exponentially. Instead, we propose an extension of bi-level optimization where we consider architecture parameters as the upper variable while weight and the associated bit-precision as a joint set of lower variables. It has been shown that a single update to both weight and quantization parameters can preserve similar performance while enjoying half the complexity of their alternating (i.e., update the weights first, then the next step involves updating bit precision parameters while keeping the weights fixed, thus doubling the update steps) updates (Cai & Vasconcelos (2020)).

We reformulate Eq. 1 to introduce the mixed-precision search weighting parameter, $\gamma$, as follows:

$$
\begin{aligned}
\min_{\alpha} \quad & \mathcal{L}_{val}(\omega^*(\alpha), \gamma^*(\alpha), \alpha) \\
\text{s.t.} \quad \omega^*(\alpha), \gamma^*(\alpha) = & \operatorname*{argmin}_{\omega, \gamma} \quad \mathcal{L}_{train}(\omega, \gamma, \alpha)
\end{aligned}
\tag{6}
$$

The exact solution to Eq. 6 involves expensive inner optimization of $\{w, \gamma\}$ over the whole training dataset. This can be avoided by approximating the gradient over a single training step (instead of the whole dataset) as follows (Liu et al. (2018b)):

$$
\begin{aligned}
\nabla_{\alpha} \mathcal{L}_{\text{val}}(w^*(\alpha), & \gamma^*(\alpha), \alpha) \\
\approx \nabla_{\alpha} \mathcal{L}_{\text{val}} \, ( & w - \xi_{\omega} \nabla_w \mathcal{L}_{\text{train}}(w, \gamma, \alpha), \\
& \gamma - \xi_{\gamma} \nabla_{\gamma} \mathcal{L}_{\text{train}}(w, \gamma, \alpha), \alpha)
\end{aligned}
\tag{7}
$$

where $\xi_{\omega}$ and $\xi_{\gamma}$ denote the learning rate of inner step optimization for weight and bit-precision, respectively. Effectively, this implies that updates on $\alpha$ and $\{w, \gamma\}$ occur in an alternating manner.

### 3.3 Complexity-aware Loss

Another challenge in joint architecture, weight, and precision search is that the existing methods predominantly rely on validation loss as its search parameter. This would always lead the $\nabla$`QDARTS` mixed-precision search to converge to a uniform bit-precision (i.e., the highest available bit-precision in the search space) for all the operations.

To overcome this, $\nabla$`QDARTS` introduces a pivotal enhancement: the integration of a complexity-aware loss function denoted in Eq. 8. This augmentation allows for a more nuanced and refined search, considering not just validation loss but also the intricate complexities inherent in the network operations. As a result, the optimization process becomes more capable of discerning varying levels of precision for different operations, leading to a more efficient search solution:

$$
L[F] = R_E(F) + \nu R_C(F)
\tag{8}
$$

where $R_E[F]$ is categorical cross-entropy loss and $R_C[F]$ represents the total computational cost of the given architecture. $R_C(F)$ is computed as follows:

$$
R_C(F) = \sum_{i=1} OPs * BitA * BitW
\tag{9}
$$

$$BitA = \gamma_i^p a_i \quad BitW = \gamma_i^q w_i \tag{10}$$

$$OPs = InChannel \times OutChannel \times (kerSize)^2$$
$$\times (\frac{InW \times InH}{stride^2}) \tag{11}$$

The hyperparameter $\nu$ determines the complexity of the final architecture. A higher $\nu$ will result in an architecture that favors lower mixed-precision architectures.

### 3.4 Increasing Model Capacity

In PC-DARTS, the training stage needs an increase in employed cells compared to the search stage to increase the capacity of the architecture to reach better accuracy. Usually, the same normal/reduction cells discovered during the search stage are stacked together. However, the straightforward stacking approach will not work in $\nabla$QDARTS due to variations in bit-precision among the discovered cells. This brings a challenge in selecting the appropriate cells for stacking, thereby increasing the capacity of the final architecture. Addressing this challenge necessitates a smart approach to identify cells that ensure optimal performance in expanding the architecture's capacity.

Through exploring various methods, we realized a technique to enhance the architecture of $\nabla$QDARTS during the training phase. By duplicating the first cells and placing them at the top of the found cells, we effectively increase the capacity of the underlying architecture. This approach is motivated by the recognition that initial layers often possess weights and activations that are more sensitive to quantization and play a significant role in the final accuracy. Leveraging this understanding, $\nabla$QDARTS strategically employs the first cell and stacks multiple copies of it on top of other discovered cells. As a result, the architecture's capability is expanded, leading to improved accuracy. We explored alternatives such as selecting cells from the last, the middle, and employing random selection, but it was the replication of the first cells that yielded the most promising results.

## 4 Experimental Setup and Evaluation

### 4.1 Setup

We have released the source code on GitHub[1]. We perform $\nabla$QDARTS experiments on CIFAR10 and ImageNet, as the two most popular datasets for evaluating the efficiency and scalability of the NAS algorithm.

We employ HWGQ (Cai et al. (2017)) as a quantization scheme for 2- and 4-bit and channel-wise min-max (Zmora et al. (2019); Gholami et al. (2021)) for 8-bit quantization. We used ffcv-based accelerator data loading (Leclerc et al. (2022)) to reduce the training time of all baselines and $\nabla$QDARTS.

Unless we explicitly mention it, we use the same setting of PC-DARTS for the DARTS-based baselines and $\nabla$QDARTS. For APQ (Wang et al. (2020)), OQA (Shen et al. (2021)), SPOS (Guo et al. (2020)), and MNAS (Gong et al. (2019)) baselines, we use the best setting introduced in the manuscripts.

The experiments are carried out on 8 (1) A40 GPUs on our internal clusters for ImageNet (CIFAR10) datasets. The training batch size is 2048(256) for ImageNet (CIFAR10) datasets. We consider 500 epochs for finetuning the searched architecture for all our experiments.

During the search phase, we use different optimizers for the $\omega$, $\alpha$, and $\gamma$. For the $\omega$, an SGD optimizer with a learning rate of 0.5 (0.1 for CIFAR10), momentum 0.9, and weight decay of $3 \times 10^{-4}$ is used. For the $\alpha$, Adam optimizer with learning rate $6 \times 10^{-3}$ ($6 \times 10^{-4}$ for CIFAR10), beta1 0.5, beta2 0.999, and weight decay of $10^{-3}$ is used. Finally, for $\gamma$, an SGD optimizer with a learning rate of 0.01, momentum of 0.9, and weight decay of $10^{-3}$ is used.

### 4.2 Baselines

Unfortunately, there is no direct SOTA work that is comparable to our work as mentioned in Table 1. However, we show how much $\nabla$QDARTS can reduce the BitOps compared to the full-precision PC-DARTS

---

[1]https://github.com/gatech-sysml/NablaQDARTS

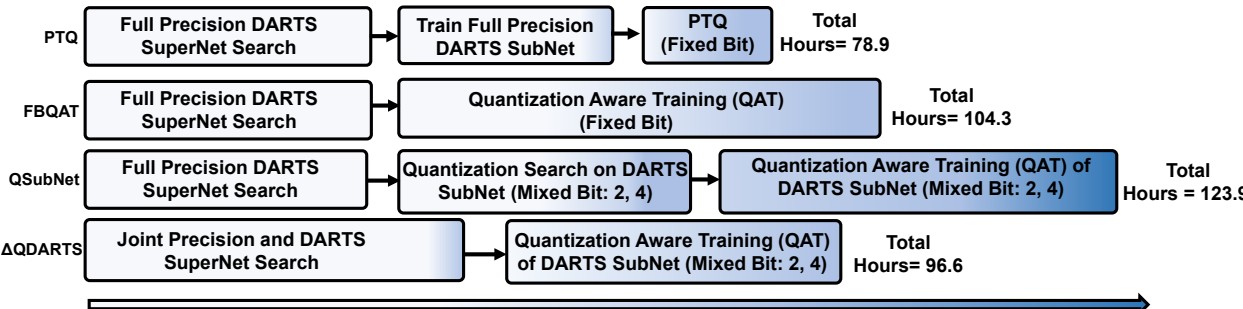

Figure 2: Comparing the total cost (both search and finetune/retrain time) of various baselines against ∇QDARTS cost to evaluate the effectiveness of ∇QDARTS. We show the total cost in terms of GPU hours for the ImageNet dataset.

(32-bit) at the cost of negligible accuracy drop. In addition, we consider assorted baselines that can be employed to be compared against ∇QDARTS as shown in Figure 2.

**Post Training Quantization (PTQ):** A simple alternative to ∇QDARTS is applying post-training quantization (PTQ) on the final trained architecture with fp32 precision (i.e., *SubNet*) discovered by PC-DARTS. This requires no additional fine-tuning and therefore is highly computationally efficient. We employ asymmetric linear post-training quantization for this purpose (Gholami et al. (2021)).

**Fixed-bit Quantized Aware Training (FBQAT):** The high accuracy drop of lower-bit quantization is usually alleviated with quantization-aware training (QAT) (Shen et al. (2021)). It is well known that quantization-aware training outperforms post-training quantization, especially in low-bit regimes (Gholami et al. (2021)). Therefore, we apply QAT to the PC-DARTS full precision searched architecture using the HWGQ (Cai et al. (2017)) technique with a uniform, fixed-bit quantization policy (e.g., three experiments with 2-, 4-, and 8-bit).

**Quantization on *SubNet* (Q*SubNet*):** To understand if ∇QDARTS is better than a multi-staged optimization that involves a NAS followed by a mixed precision architecture search, we apply differentiable mixed-precision quantization technique (i.e., EdMIPS) on the output of the search stage of PC-DARTS. We call this baseline Q*SubNet*. After 50 epochs of EdMIPS search on the architecture produced by PC-DARTS, we apply 450 epochs of fine-tuning so that it is a fair comparison against ∇QDARTS.

**Existing Baselines:** From the literature, we compare against APQ (Wang et al. (2020)), OQA (Shen et al. (2021)), SPOS (Guo et al. (2020)), and MNAS (Gong et al. (2019)) as the closest work to ∇QDARTS. We also compare ∇QDARTS with state-of-the-art non-NAS methods such as EMQ (Dong et al. (2023)), HAWQ (Yao et al. (2021)), OMPQ (Ma et al. (2023)).

### 4.3 Accuracy/BitOps/Memory Footprint Results

Table 2 and Table 3 compare accuracy, required bit operations (BitOps), and memory footprint of different baselines with ∇QDARTS. The first column shows the employed method. The number of employed cells in the search and training stages is shown in the second and third columns. The fourth column displays bit-precision, which pertains to both weight and activation, unless explicitly stated otherwise. The accuracy results (Top-1 for ImageNet) are shown in the fifth column. The sixth and seventh columns show the improvement in terms of bit operations normalized to the PC-DARTS values. The last column indicates the amount of improvement in memory footprint normalized to PC-DARTS. The absolute values of PC-DARTS are shown in parentheses in the first row (i.e., in terms of Gigabits for the required bit operations and megabytes (MB) for memory footprint). The difference between the two versions of ∇QDARTS (2,4) is the complexity-aware loss values. By changing this parameter ($\nu$ in Eqn.8), we can push for higher accuracy at the cost of higher BitOps.

### 4.4 CIFAR10 Results

As demonstrated in Table 2, the effectiveness of ∇QDARTS shines for CIFAR10 dataset. With a mere utilization of 2- and 4-bit operations, ∇QDARTS manages to achieve remarkable reductions in required bit operations by a

Table 2: Comparing the impact of different methods on Accuracy, BitOps, and Memory Footprint over CIFAR10 dataset

| Method | #Cells-search | #Cells-train | Precision | Accuracy | BitOps (G) | Memory Footprint (MB) |
|---|---|---|---|---|---|---|
| PC-DARTS | 8 | 20 | 32 | 96.25% | 1(634.712) | 1(218.977) |
| PTQ | 8 | 20 | 8 | 95.59% | 0.0625 | 0.250 |
| PTQ | 8 | 20 | 4 | 18.19% | 0.0150 | 0.125 |
| PTQ | 8 | 20 | 2 | 9.99% | 0.0039 | 0.063 |
| FBQAT | 8 | 20 | 8 | 95.86% | 0.0611 | 0.236 |
| FBQAT | 8 | 20 | 4 | 62.68% | 0.0157 | 0.125 |
| FBQAT | 8 | 20 | 2 | 38.45% | 0.0039 | 0.063 |
| $QSubNet$ | 8 | 20 | 2, 4 | 70.05% | 0.0039 | 0.063 |
| $\nabla$QDARTS | 8 | 8 | 2,4 | 93.43% | **0.0056** | **0.047** |
| $\nabla$QDARTS | 8 | 8 | 2,4 | **94.68**% | 0.0062 | 0.062 |
| $\nabla$QDARTS | 8 | 8 | 2,4,8 | 95.95% | 0.0185 | 0.081 |
| $\nabla$QDARTS | 8 | 20 | 2, 4, 8 | **96.20**% | 0.0556 | 0.198 |

staggering $160\times$, along with substantial savings in memory footprint of $16\times$ while the accuracy drop is $1.57\%$. $\nabla$QDARTS showcases its capacity to attain the same level of accuracy as PC-DARTS while simultaneously reducing the required operations by $18\times$ and the memory footprint by $5\times$.

As can be observed, $\nabla$QDARTS achieves remarkable accuracy in CIFAR10 without having to increase training capacity as compared to search capacity. (i.e., merely $0.3\%$ lower accuracy than fp32 with PC-DARTS). This exemplifies the efficiency of $\nabla$QDARTS in determining the optimal architecture and weight configurations, which enables us to reach high accuracy with less capacity. The last row in Table 2 shows that $\nabla$QDARTS reaches the same accuracy as PC-DARTS with $18\times$ less BitOps and $5\times$ less memory footprint.

Comparing against the best outcomes achieved by other baselines (i.e., $QSubNet$), it is worth noting that $\nabla$QDARTS excels. With $1.58\times$ the bit operations required by $QSubNet$, $\nabla$QDARTS improves the accuracy by an impressive $24.63\%$ while outperforming $QSubNet$ in terms of consumed GPU hours by $28\%$. This further highlights the superiority of $\nabla$QDARTS in optimizing accuracy while maximizing efficiency in comparison to other baselines.

## 4.5 ImageNet Results

As shown in Table 3, for the ImageNet dataset, we observe a significant accuracy degradation (i.e., $25.1\%$) for the $QSubNet$ when utilizing 2- and 4-bit operations compared to PC-DARTS. In contrast, $\nabla$QDARTS, with $1.5\times$ less required BitOps, reaches $21\%$ higher accuracy than $QSubNet$. Although $\nabla$QDARTS with only 2- and 4-bit precision for both activation and weights experiences an accuracy drop of $1.7\%$ compared to PC-DARTS, it reduces the required bit operations by an outstanding $65.36\times$ and the memory footprint by $5\times$.

Moreover, by incorporating 2-, 4-, and 8-bit operations, $\nabla$QDARTS further mitigates the accuracy drop to a mere $1\%$. Particularly, this is achieved by reducing the required bit operations to a notable $17\times$, underscoring the superior performance of $\nabla$QDARTS in optimizing accuracy and reducing computational requirements on the ImageNet dataset.

### 4.5.1 Compare with SOTA

We also compare $\nabla$QDARTS with APQ (Wang et al. (2020)), MNAS (Gong et al. (2019)), OQA (Shen et al. (2021)), and SPOS (Guo et al. (2020)) as the closest state-of-the-art baselines to $\nabla$QDARTS as shown in Table 3. APQ, SPOS, OQA, and MNAS converge at the accuracy of $72.1\%$, $71.5\%$, $74.1\%$, and $71.77\%$ while APQ and MNAS methods use even 8-bit weight/activation. In contrast, by only using 2- and 4-bit, $\nabla$QDARTS outperforms APQ, SPOS, OQA, and MNAS by $2.3\%$, $2.9\%$, $0.3\%$, and $2.7\%$ in terms of accuracy. By involving 8-bit precision, $\nabla$QDARTS outclasses APQ, SPOS, OQA, and MNAS by $3\%$, $3.6\%$, $1\%$, and $3.4\%$, respectively. In terms of BitOps (memory footprint), by considering almost the same accuracy, $\nabla$QDARTS outperforms APQ, SPOS, OQA, and MNAS by $2.3\times$ $(12\times)$, $2.4\times(3\times)$, $13\%(6.2\times)$, $3.4\times(37\%)$.

Figure 3 illustrates the accuracy-BitOps trade-off for different versions of $\nabla$QDARTS compared to other state-of-the-art methods. As shown, $\nabla$QDARTS effectively pushes the frontier toward the top-left, achieving higher accuracy with fewer BitOps.

Table 3: Comparing the impact of different methods on Accuracy, BitOps, and Memory Footprint over ImageNet dataset

| Method | #Cells-search | #Cells-train | Precision | Accuracy | BitOps (G) | Memory Footprint (MB) |
|---|---|---|---|---|---|---|
| PC-DARTS | 8 | 14 | 32 | 76.1% | 1(536.552) | 1(172.102) |
| PTQ | 8 | 14 | 8 | 67.47% | 0.0625 | 0.254 |
| PTQ | 8 | 14 | 4 | 1.0% | 0.0157 | 0.127 |
| FBQAT | 8 | 14 | 8 | 74.01% | 0.0622 | 0.244 |
| FBQAT | 8 | 14 | 4 | 50.27% | 0.0156 | 0.127 |
| FBQAT | 8 | 14 | 2 | 33.57% | 0.0039 | 0.063 |
| $QSubNet$ | 8 | 14 | 2,4 | 50.70% | 0.0156 | 0.127 |
| APQ | (21 blocks) | (21 blocks) | 4,6,8 | 72.1% | 0.0239 | 1.284 |
| MNAS | (22 blocks) | (22 blocks) | 2,4,6,8 | 71.77% | 0.0347 | 0.147 |
| OQA | (21 blocks) | (21 blocks) | 4 | 74.1% | 0.0173 | 1.245 |
| SPOS | (20 blocks) | (20 blocks) | 1,2,3,4 | 71.5% | 0.0244 | 0.321 |
| ∇QDARTS | 8 | 14 | 2,4 | 71.70% | **0.0101** | **0.107** |
| ∇QDARTS | 8 | 14 | 2,4 | **74.41**% | 0.0153 | 0.199 |
| ∇QDARTS | 8 | 14 | 2,4,8 | **75.10**% | 0.0587 | 0.385 |

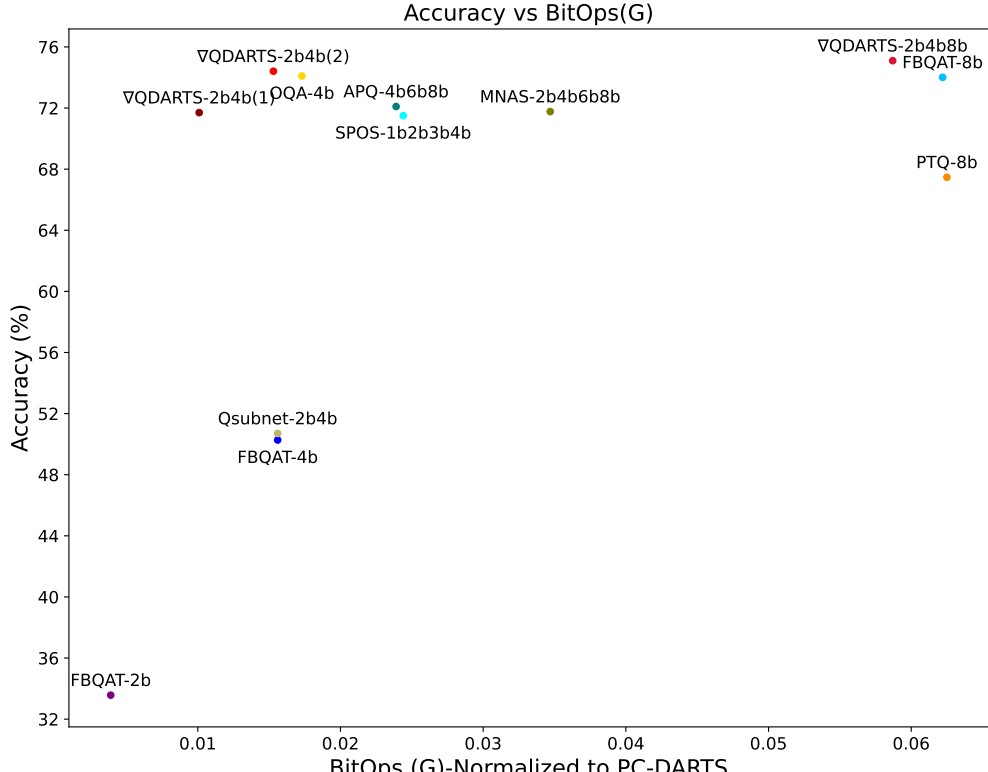

Figure 3: Comparing accuracy-BitOps trade-off of differentiae version of ∇**QDARTS** and SOTA methods over ImageNet. ∇**QDARTS** outperforms existing approaches by providing higher accuracy and/or lower BitOps.

## 4.6 Total Cost

Figure 2 and Table 4 offer a comprehensive depiction of the total needed time (i.e., search and training), measured in GPU hours, for various methods on the ImageNet dataset on 8 A40 GPUs. Upon observation, it is evident that ∇**QDARTS** takes 28% (8%) shorter runtime compared to $QSubNet$ (FBQAT), and longer runtime compared to PTQ. PTQ takes less time since it assumes that an expensive, fully trained network is available to be quantized with fixed-bit and no extra fine-tuning is required. Compared with state-of-the-art methods ∇**QDARTS** improves the total cost significantly. ∇**QDARTS** improves the total search and training cost by 3.1×, 1.54×, compared with APQ, OQA, respectively. SPOS has less GPU hours at the cost of low accuracy, high BitOps, and a large memory footprint.

Table 4: Comparing the total GPU hours (8,A40) of ∇QDARTS and other methods on ImageNet dataset

| Method | APQ Wang et al. (2020) | OQA Shen et al. (2021) | SPOS Guo et al. (2020) | ∇QDARTS |
|---|---|---|---|---|
| GPU Hours | 302.6 | 149.4 | 73.5 | 96.6 |

## 4.7 Compare with Non-NAS Quantization Methods

We compare ∇QDARTS with non-NAS state-of-the-art quantization schemes in Table 5. The numbers are the highest accuracy that these methods can reach with their corresponding BitOps(i.e., normalized to PC-DARTS). Among them, only HAWQ supports mixed-precision for both weight and activation.

In terms of accuracy, OMPQ, EMQ, and HAQW3 on ResNet50 have 1.18% (1.87%), 1.6% (2.3%), and 0.29% (0.89%) higher accuracy compared to two versions of ∇QDARTS with (2,4) bit setting. However, their BitOps are 495% (1900%), 460% (1800%), and 489% (1870%) higher than ∇QDARTS (2,4). Notably, PCDARTS in FP32 demonstrates lower accuracy compared to quantized ResNet50 (76.1% vs. 76.7%). This indicates that the higher accuracy of these methods is attributed to the inherent capacity of ResNet50 relative to PCDARTS. However, by increasing the number of cells in PCDARTS, it can reach higher accuracy, at the expense of increased BitOps. Within the same BitOps budget, ∇QDARTS outperforms all these methods in terms of accuracy.

Table 5: Comparing ∇QDARTS with non-NAS quantization schemes on ImageNet dataset

| Method | W/A | Accuracy | BitOps(G) |
|---|---|---|---|
| OMPQ-ResNet18 | 4-8/8 | 72.30 | 0.1808 |
| OMPQ-ResNet18 | 4-8/6 | 72.08 | 0.1398 |
| OMPQ-ResNet50 | 4-8/5 | **76.28** | 0.2907 |
| HAWQv3-ResNet18 | 4,8/4,8 | 70.22 | 0.1342 |
| HAWQv3-ResNet18 | 4,8/4,8 | 71.56 | 0.2162 |
| HAWQv3-ResNet50 | 4,8/4,8 | **75.39** | 0.2870 |
| EMQ–ResNet18 | 2,3,4/8 | 72.31 | 0.1715 |
| EMQ–ResNet18 | 2,3,4/6 | 72.28 | 0.1323 |
| EMQ–ResNet50 | 2,3,4/5 | **76.70** | 0.2758 |

## 4.8 Ablation Study

In this subsection, we study different aspects of the proposed ∇QDARTS and perform different experiments to show the impact of various parameters on the proposed methods.

### 4.8.1 Alternative Update Methods

In the DARTS framework, bi-level optimization concurrently optimizes network weights and architecture parameters using gradient descent. The upper-level optimization targets architecture parameters, whereas the lower level focuses on network weights. Conversely, one-level optimization merges these stages, simplifying the process at the potential expense of architecture quality. In ∇QDARTS, the complexity escalates as it involves three optimization parameters: architecture parameters ($\alpha$), weights ($w$), and bit-precision ($\gamma$).

As discussed in the main paper, incorporating $\gamma$ introduces a tri-level optimization challenge, substantially complicating the optimization process. Thus, we propose an efficient bi-level optimization variant in ∇QDARTS.

Below, we outline various update mechanisms and their effects on accuracy and BitOps.

∇QDARTS **(Efficient Bi-level Optimization):** In ∇QDARTS, the sequence involves forward propagation with a validation dataset, backpropagation on $\alpha$, a forward pass with the training dataset, and subsequent backpropagation for $w$ and $\gamma$.

$$
\min_{\alpha} \quad \mathcal{L}_{\text{val}}(\omega^*(\alpha), \gamma^*(\alpha), \alpha)
$$
$$
\text{s.t.} \quad \omega^*(\alpha), \gamma^*(\alpha) = \operatorname*{argmin}_{\omega, \gamma} \quad \mathcal{L}_{\text{train}}(\omega, \gamma, \alpha) \tag{12}
$$

**Bi-level-other Optimization:** This optimization method (Eq. 13) starts with a forward pass using a validation dataset, followed by backpropagation for $\alpha$ and $\gamma$. Then, a forward pass with the training dataset precedes backpropagation for $w$.

$$\min_{\alpha,\gamma} \quad \mathcal{L}_{\text{val}}(\omega^*(\alpha,\gamma),\gamma,\alpha)$$
$$\text{s.t.} \quad \omega^*(\alpha,\gamma) = \operatorname*{argmin}_{\omega} \quad \mathcal{L}_{\text{train}}(\omega,\gamma,\alpha) \tag{13}$$

**Tri-level Optimization:** This optimization scenario (Eq. 14) begins with a forward pass using a validation dataset and backpropagation for $\alpha$. Next, a forward pass with the training dataset is followed by backpropagation for $w$. Finally, another forward pass with the training set leads to backpropagation for $\gamma$.

$$\min_{\alpha} \quad \mathcal{L}_{\text{val}}(\omega^*(\alpha,\gamma),\gamma,\alpha)$$
$$\text{s.t.} \quad \omega^*(\alpha,\gamma) = \operatorname*{argmin}_{\omega} \quad \mathcal{L}_{\text{train}}(\omega,\gamma^*(\omega),\alpha) \tag{14}$$
$$\text{s.t.} \quad \gamma^*(\omega) = \operatorname*{argmin}_{\gamma} \quad \mathcal{L}_{\text{train}}(\omega,\gamma,\alpha)$$

**One-level Optimization:** This optimization procedure (i.e., Eq. 15) eliminates the need for a validation dataset, conducting a forward pass with the training dataset and then backpropagating for $w$, $\alpha$, and $\gamma$.

$$\min_{\alpha,\omega,\gamma} \quad \mathcal{L}_{\text{train}}(\omega,\gamma,\alpha) \tag{15}$$

Figure 4 illustrates the variations in accuracy and BitOPs across different optimization methods where all of them use the same hyperparameter. Each label indicates optimization method, BitOps, and accuracy. Figure 4 shows that while alternative methods reduce the accuracy, $\nabla$`QDARTS` maintains higher accuracy with fewer BitOps. Table 6 compares the search time of optimization methods. The One-level needs only one forward pass with the training dataset per mini-batch, per epoch. Hence, it gives the lowest search time and lowest accuracy. Compared to Tri-Level and Bi-Level-other, $\nabla$`QDARTS` improve the search time by 41.8% and 6.6%, respectively. Surely, for the larger datsets, we will see more improvement. For example, for the ImageNet dataset, we observe a timeout issue for Tri-Level optimization (5 days).

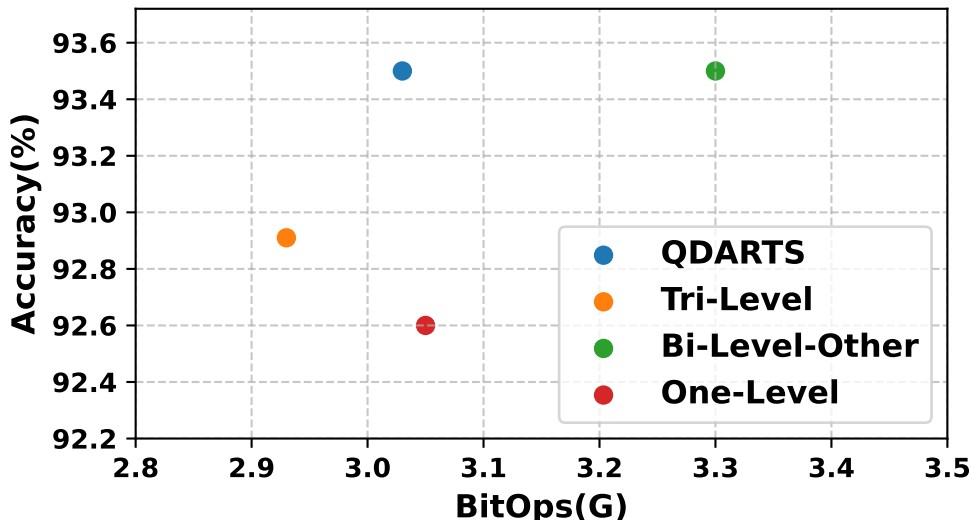

Figure 4: Comparing the impact of various optimization methods on accuracy and BitOps.

Table 6: Comparing search time (minutes) of various optimization methods over CIFAR10 dataset

| $\nabla$`QDARTS` | Bi-Level-other | Tri-Level | One-Level |
|---|---|---|---|
| 372.28 | 398.83 | 640.02 | 272.08 |

### 4.8.2 Validation Accuracy

Figure 5 illustrates how the validation accuracy increases through the first 250 training epochs over the ImageNet dataset. As it has shown, Q*SubNet* and FBQAT (4-bit) achieve almost similar performance and convergence rate throughout the training period and they are significantly better than FBQAT (2-bit). $\nabla$QDARTS outperforms all the baselines extensively from the beginning of the training.

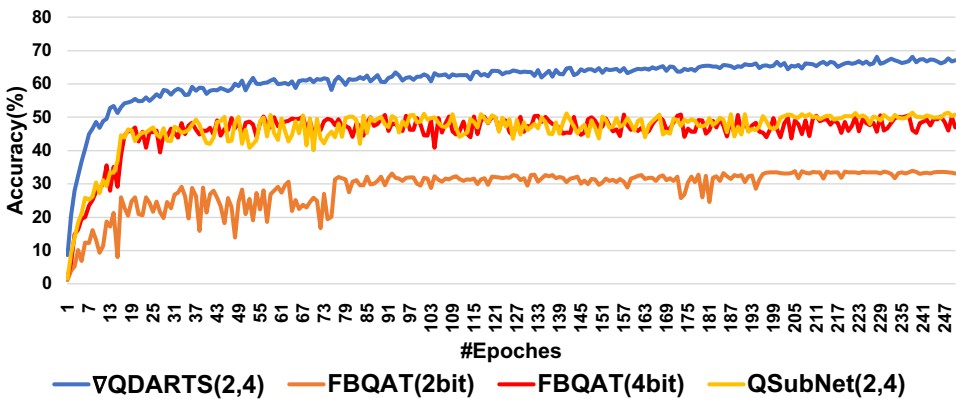

Figure 5: Validation accuracy for different techniques throughout the first 250 training epochs (ImageNet).

### 4.8.3 Required Bit Operations Per Cell

Figure 6 compares the required bit operations of different cells of $\nabla$QDARTS that have 2-,4-bit, and also $\nabla$QDARTS with 2-, 4-, 8-bit against PC-DARTS. The Cell#5 and Cell#8 are *reduction cells* and the rest are *normal cells.*

The nuanced dissimilarities observed among different normal cells in PC-DARTS can be attributed to a fundamental distinction between cells positioned directly after a *normal cell* and those following a *reduction cell.* Notably, the cells immediately succeeding a reduction cell lead to variations in the computational requirements across normal cells in PC-DARTS. This complex architectural style highlights the detailed design decisions made to improve performance. In the case of $\nabla$QDARTS, the efficacy is exemplified by the substantial reduction in required operations for each cell. Specifically, $\nabla$QDARTS achieves a significant reduction in bit operations of up to $65.5\times$ for the (2,4) bit configuration and a $17\times$ for the (2,4,8) bit configuration. These reductions further emphasize the effective computational optimization achieved by $\nabla$QDARTS.

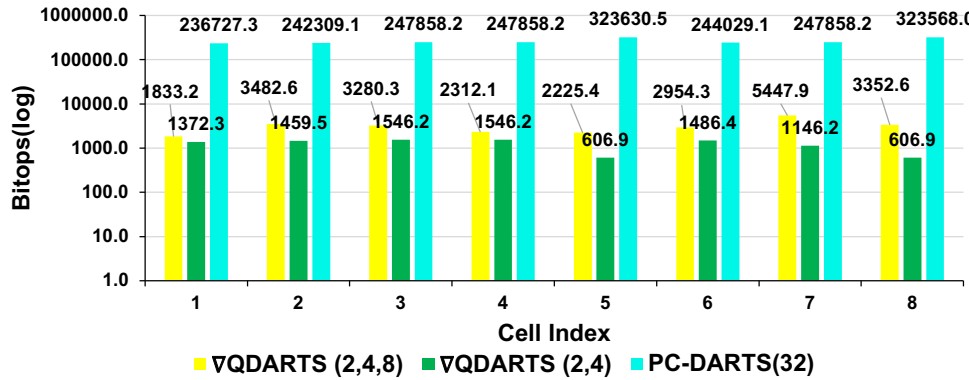

Figure 6: Comparing required bit operations of different cells for PC-DARTS and $\nabla$QDARTS (ImageNet).

### 4.8.4 Sensitivity Analysis of Complexity Decay Parameter ($\nu$)

In this analysis, we explore the effects of altering the complexity decay parameter ($\nu$) within the context of complexity-aware loss on the resulting accuracy. Previously, in Table 2 and Table 3 of the main paper, we present diverse accuracy/BitOps metrics for the same bit precision levels within the $\nabla$QDARTS framework, highlighting the sole variable as the complexity decay parameter ($\nu$). Here we extend this experiment. As shown in Table 7, altering the complexity decay parameter by a factor of $10^{10}$ results in a 0.58% change in accuracy on the CIFAR10 dataset. However, its impact on the ImageNet dataset is significant. By changing this hyperparameter from $10^{-4}$ to $10^{-9}$, the accuracy varies by 6.81%.

Table 7: Comparing the impact of different values of complexity decay ($\nu$) on accuracy over CIFAR10 and ImageNet datasets

| Method | #Cells-search | #Cells-train | Precision | Complexity decay | Accuracy | Dataset |
|---|---|---|---|---|---|---|
| $\nabla$QDARTS | 8 | 8 | 2,4 | $10^{-5}$ | 94.28 % | CIFAR10 |
| $\nabla$QDARTS | 8 | 8 | 2,4 | $10^{-7}$ | 94.43% | CIFAR10 |
| $\nabla$QDARTS | 8 | 8 | 2,4 | $10^{-9}$ | 94.49% | CIFAR10 |
| $\nabla$QDARTS | 8 | 8 | 2,4 | $10^{-15}$ | **94.86**% | CIFAR10 |
| $\nabla$QDARTS | 8 | 14 | 2,4 | $10^{-4}$ | 64.45% | ImageNet |
| $\nabla$QDARTS | 8 | 14 | 2,4 | $10^{-9}$ | **71.26**% | ImageNet |

Employing Discovered Precision of the first/last Cell for Stacking Cells During Retraining Phase: As we mentioned in Section 3.4, increasing the model capacity in $\nabla$QDARTS is not straightforward as different cells have different precision. Table 8 shows how stacking the cells using the discovered cells will impact the accuracy. As seen, by replicating the first cells and positioning them at the top of existing cells, we enhance the capacity of the underlying architecture compared to using the cells discovered at the end.

Table 8: Study the impact of employing the discovered precision of the first cell and the last cell for stacking cells during the retraining phase on accuracy over ImageNet dataset

| Method | #Cells-search | #Cells-train | Precision | Position | Accuracy |
|---|---|---|---|---|---|
| $\nabla$QDARTS | 8 | 14 | 2,4,8 | last | 74.608% |
| $\nabla$QDARTS | 8 | 14 | 2,4,8 | first | 75.04 % |

In the appendices, we have provided more ablation studies to show how the bit-precision parameter ($\gamma$) evolved during the search phase. Furthermore, we study the impact of the complexity decay parameter, having additional cells in the training phase.

## 5 Conclusion

The $\nabla$QDARTS method represents a significant advancement in architecture-weight-precision joint search, skillfully integrating weight values, architecture, and bit-precision (for both weight and activation) search into a single-shot and differentiable framework. $\nabla$QDARTS approach not only eliminates the need for proxy and pretraining but also demonstrates its effectiveness through a notable enhancement in accuracy, reducing required bit operations and memory footprint. $\nabla$QDARTS achieves remarkable results on CIFAR10 using (2,4) bit precision compared to fp32, cutting bit operations by 160× while only decreasing accuracy by 1.57%. Enhancing the capacity allows $\nabla$QDARTS to equal fp32 accuracy, decreasing bit operations by 18×. Using (2,4,8) bit precision, $\nabla$QDARTS reduces the accuracy drop to 1% relative to fp32, while significantly cutting bit operations by 17× and memory footprint by 2.6×. Regarding bit operation (memory footprint), $\nabla$QDARTS surpasses APQ, SPOS, OQA, and MNAS with comparable accuracy by APQ, SPOS, OQA, and MNAS with similar accuracy by 2.3× (12×), 2.4× (3×), 13% (6.2×), 3.4× (37%), respectively. $\nabla$QDARTS improves the total search and training cost by 3.1×, 1.54×, compared with APQ, OQA, respectively. These pieces of evidence show that $\nabla$QDARTS holds promise for efficient and powerful network design, showcasing the potential of joint quantization with NAS in optimizing computational resources while maintaining high accuracy.

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

# A  Appendix

## A.1  Visual Analysis of Bit Precision and Architecture Identified by ∇QDARTS

The plots in Fig. 9 showcase the structure of the discovered cells within ∇**QDARTS** when employing a (2,4,8) bit configuration for the ImageNet dataset, juxtaposed against PC-DARTS (i.e., first row). A noteworthy observation within the visualization is the strategic selection of bit precision for various convolution blocks. Notably, 2-bit precision is predominantly chosen for many of the 5x5 convolution blocks, while a higher precision of either 4 or 8 bits is judiciously employed for the 3x3 convolution blocks. This representation demonstrates ∇**QDARTS**' ability to uncover computationally efficient operations for both weight and activation within each cell, attesting to its optimization capabilities. Figures 7 and 8 show detailed analysis of which operations tend to use which precision for the discovered architecture in Figure 9 for both weight and activation, respectively. In the label of each bar, the first number shows the cell number and the second number shows the layer number. As shown, a significant portion of layers (about 54%) employ 2-bit weights, indicating heavy use of ultra-low precision weights in the network. Roughly, 32% of layers use 4-bit weights, and 14% use 8-bit weights. For the activation, we observe the same trends. The contributions of 2,4, and 8 bits are 52%, 26%, and 22%. Usually 2-bit precision is chosen for bigger kernels, and 8-bit precision is selected for skip connections. It is noteworthy that the other layers in cells 2 and 5 are pooling layers, which we do not quantize them.

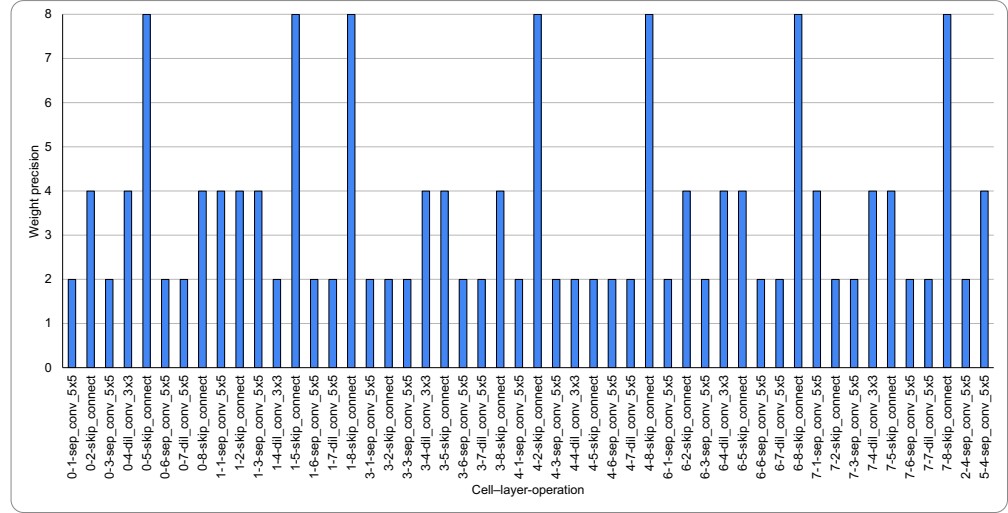

Figure 7: Operation-wise precision analysis of weights for discovered architecture in Figure 9.

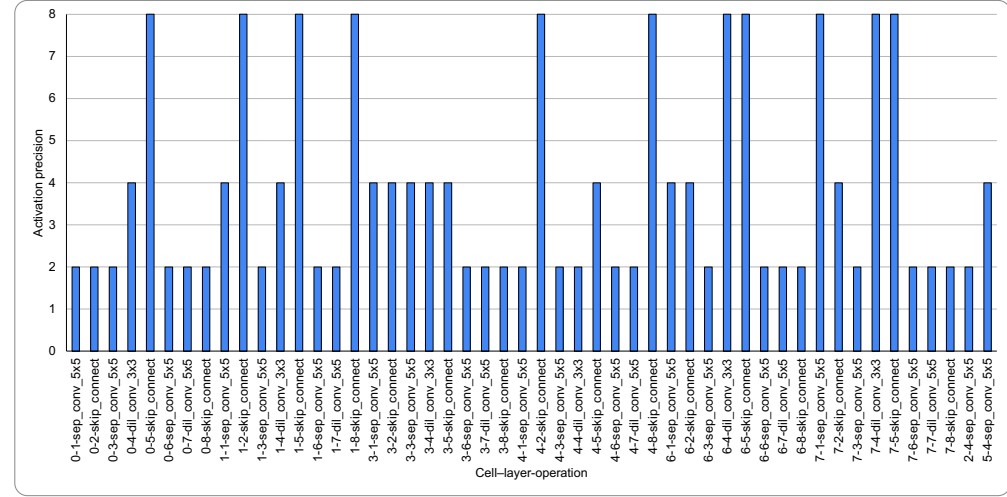

Figure 8: Operation-wise precision analysis of activation for discovered architecture in Figure 9

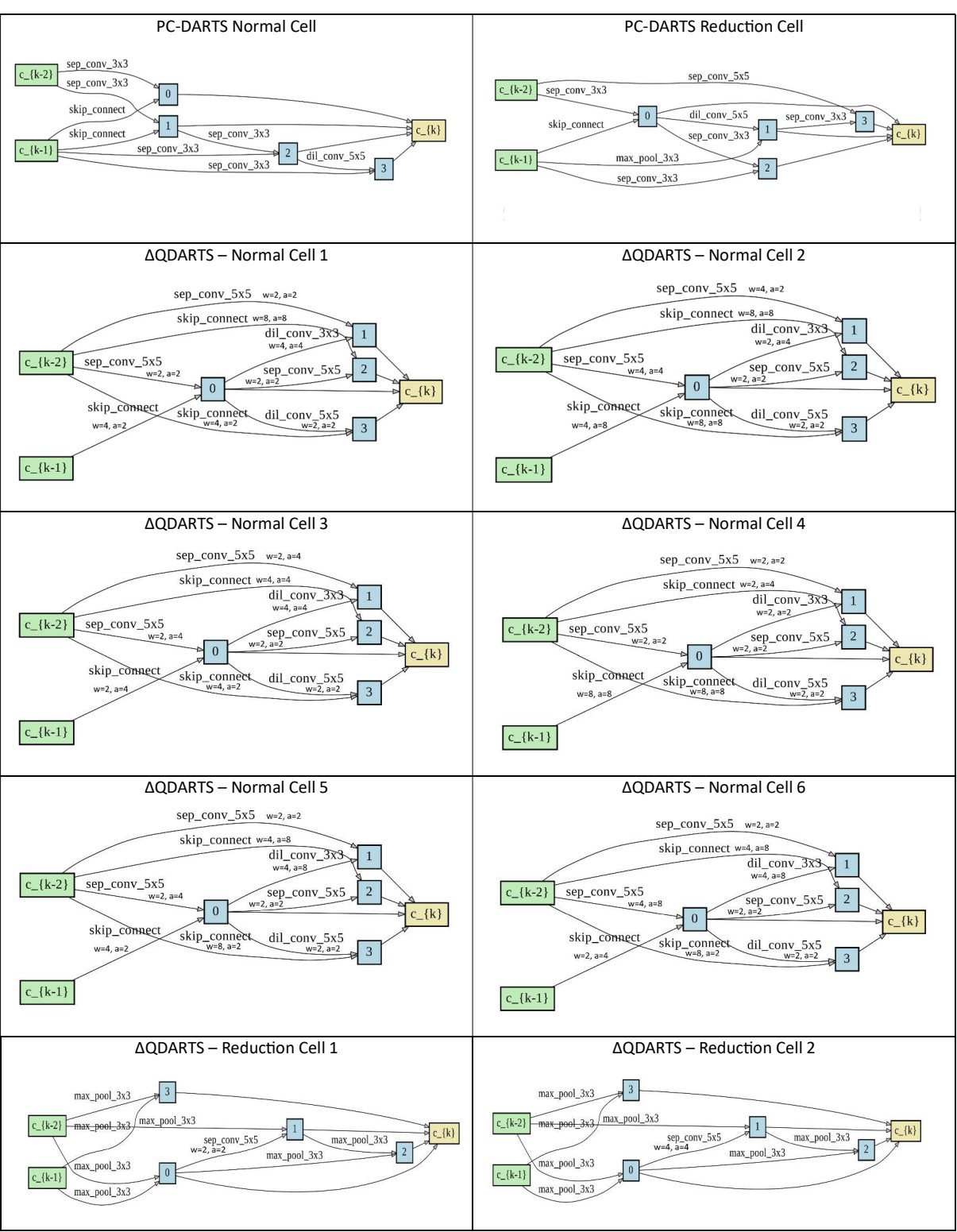

Figure 9: Comparing the discovered cell structures with bit precision of all normal and reduction cells (i.e., row 2-5) with PC-DARTS (i.e., row 1) for the ImageNet dataset. Due to mixed-precision, unlike PC-DARTS, the cells in ∇QDARTS are not similar.

## A.2 $\gamma$ Evolution During Search

The plots in Figure 10 show the evolution of the bit precision ($\gamma$) parameter over 50 epochs during the search phase for activation (first six plots) and weights (second six plots) on randomly selected cells, using the CIFAR-10 dataset with bit precisions (2, 4, 8). The Y-axis shows the probability of selecting each bit precision. The results demonstrate that the proposed framework tends to favor higher precision to enhance network capacity and improve accuracy.

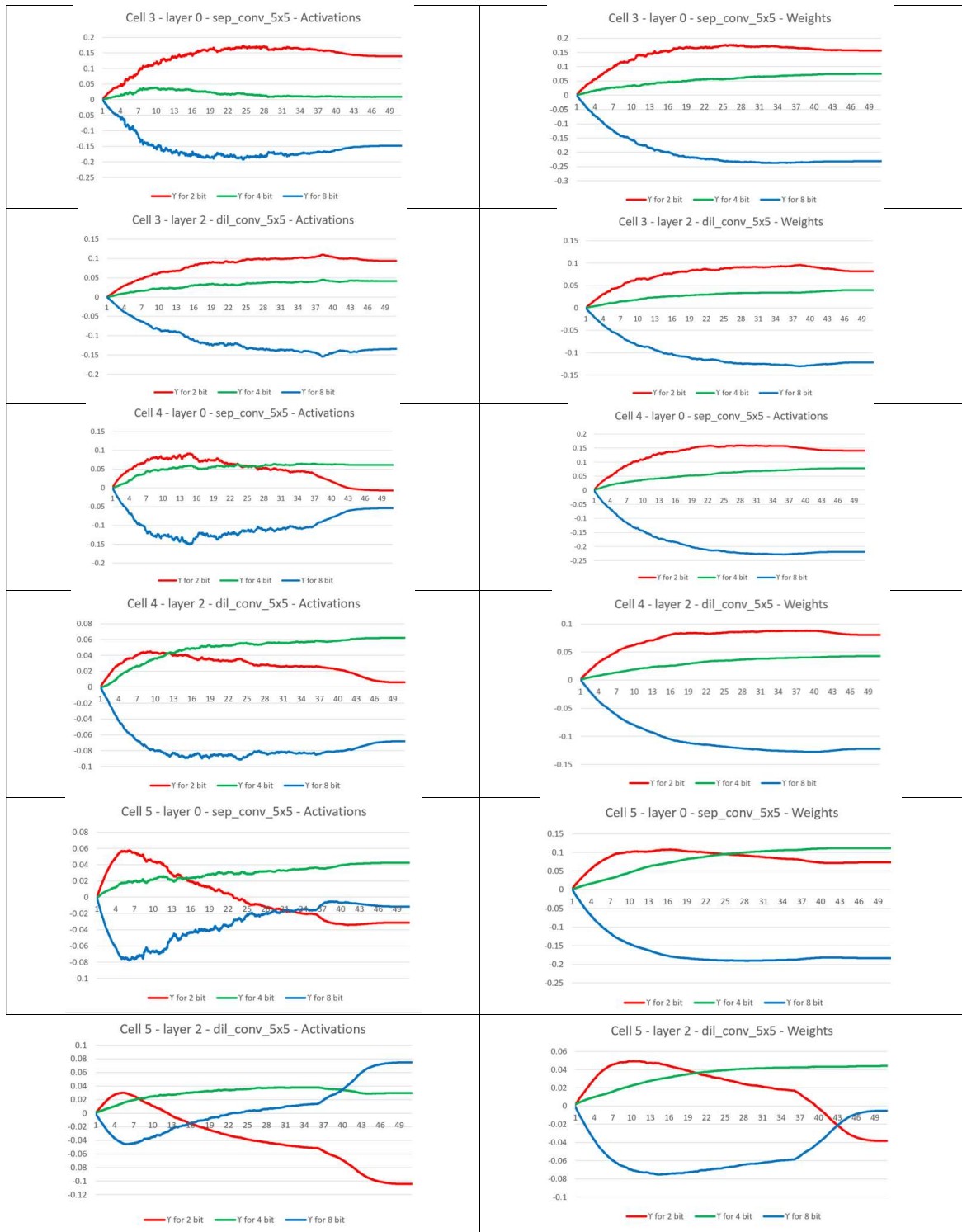

Figure 10: Evolution of the bit precision parameter.

### A.3 Latency and Energy Measurements

Most of the existing work that study quantization implement their methods using ML frameworks (e.g., PyTorch) and show the efficiency of their methods with simulation or proxy metrics rather than demonstrating actual inference on real hardware. Nevertheless, we employ two approaches to approximate how $\nabla$QDARTS can impact latency and energy on hardware.

#### First Approach: Simulation & Analysis

For the simulation-based results, we employed Timeloop (i.e, DNN accelerator mapper/simulator) (Parashar et al. (2019)) and Accelergy (i.e., energy estimator) (Wu et al. (2019b)) to evaluate latency/energy of the $\nabla$QDARTS networks and compare them with the PC-DARTS baseline in fp32 precision.

In our analysis, we use Google's Coral Edge TPU– a mobile edge AI accelerator with publicly available specifications (Coral (2020); Google Coral (2024); Medium.com (2020)). The Edge TPU was chosen for its representative mobile-edge performance and its well-documented specifications, making it suitable to configure Timeloop/Accelergy tools.

**Simulation Configuration:** We set the clock frequency to 500 MHz and configure the Processing Elements (PE) array to match the Edge TPU's compute capabilities. We assume 64 $\times$64 MAC units (4096 MACs total). At 500 MHz, 4096 MACs can perform almost 4 TOP/s. We set a scratchpad memory to 8 MB and the data width to 1 byte per element (since it stores 8-bit data primarily). Reading data from an 8MB on-chip SRAM costs around 8 pJ per 8-bit word (Balasubramonian et al. (2017); Chen (2019); Pawlowski (2019)). Since model footprints exceed 8MB, we include a DRAM level for main memory. We assume 320 pJ per 8-bit access for DRAM (Balasubramonian et al. (2017); Chen (2019); Pawlowski (2019)). We ensure the on-chip SRAM and PEs have sufficient bandwidth. E.g., the 8MB may be divided into banks.

The granularity of the precision is per layer, as mentioned in Figure 9, and we already know which operations work at what precision. We also use 0.5 pJ, 0.125 pJ, 0.03 pJ, 2 pJ, and 8 pJ for MAC-8bit, MAC-4bit, MAC-2bit, MAC-16bit, and MAC-32bit, respectively (Han et al. (2016); Zhao et al. (2025)).

#### Second Approach: FPGA Implementation

For FPGA implementation, we use AMD Xilinx Alveo U250 FPGA (AMD (2023)) that supports AMD-Xilinx's FINN framework for low-bit neural networks via LUT-based logic for binary/ternary ops (Xilinx-AMD (2023)). To keep comparisons fair with the simulation, we assume the same technology node and frequency and similar memory hierarchy constraints as the Timeloop/Accelergy study.

**FPGA Configuration:** FINN (Xilinx-AMD (2023)) yields a pipelined design where each layer processes data in a streaming fashion. The FINN accelerator would be structured to read weights from DDR, buffer them on chip, and stream the input through the pipeline. Each weight is reused for all necessary MACs (e.g., in convolutions) before loading new weights, minimizing redundant transfers.

Because FINN allows increasing parallelism when quantization reduces resource usage, the low-precision models can be significantly unrolled, reducing cycles. In contrast, the FP32 model demands more resources per operation, forcing more pipeline stages, and higher cycle count, demanding more weights/feature maps stream from off-chip DDR4. Power is estimated based on device utilization and toggling activity. Lower-bit operations switch fewer transistors per compute. Quantized circuits consume less dynamic power per operation (Ducasse et al. (2021)). FINN's post-implementation reports can provide an estimate of dynamic power given the switching rates for LUTs, DSPs, and clocks. Then, energy is computed as *Power × Latency* (Rao (2023); Venkataramaniah (2022); DeepBlue (2023)).

#### Comparing FPGA and Timeloop/Accelergy (ASIC) Results:

Table 9 summarizes the latency, and energy results for the three DNN models on the FPGA (FINN) and the ASIC (Timeloop/Accelergy). All results are for a single inference (batch=1).

Table 9: Estimated Latency and Energy Inference for Edge FPGA and ASIC under $\nabla$`QDARTS`

| Model (Gbit-ops, Memory) | Platform | Latency (ms) | Compute Energy (mJ) | Memory Energy (mJ) | Total Energy (mJ) |
|---|---|---|---|---|---|
| FP32 Model (BitOps(G) = 536.6) | FPGA | 60.7 | 8.1 | 28.3 | 36.4 |
| (Accuracy: 76.1%) | ASIC | 30.7 | 2.2 | 10.7 | 12.9 |
| Mixed 2/4/8-bit Model (BitOps(G) = 31.5) | FPGA | 15.6 | 1.3 | 5.1 | 6.4 |
| (Accuracy: 75.1%) | ASIC | 10.1 | 0.5 | 2.0 | 2.5 |
| Mixed 2/4-bit Model (BitOps(G) = 5.42) | FPGA | 8.4 | 0.2 | 1.8 | 2.0 |
| (Accuracy: 71.7%) | ASIC | 5.2 | 0.15 | 0.6 | 0.75 |

The FP32 design uses large bit-widths and drives off-chip DDR, approaching a high power draw, whereas the quantized designs use fewer bits, fewer bit toggles, and skip most off-chip memory, leading to lower power and latency. The mixed-precision models are far less demanding. The smallest 2/4-bit network executes in only a few milliseconds, with the ASIC drawing only a few hundred milliwatts (energy 0.75 mJ). This aligns with expectations that $\nabla$`QDARTS` quantization shrinks the workload by an order of magnitude, yielding lower total latency and energy versus the FP32 baseline.

Please note that the degree of $\nabla$`QDARTS` improvement can vary based on the specific hardware device selected and its associated configuration parameters. Nevertheless, adopting $\nabla$`QDARTS` consistently provides substantial advantages in terms of reduced energy consumption and lower latency, highlighting their potential for impact across various deployment scenarios.

## A.4 Approximate Complexity Decay Parameter Tuning by Data & Model

In all datasets, by adjusting the value of the complexity decay parameter ($\nu$), we can either achieve the highest accuracy (at $\nu = 0$) or minimize BitOps (at $\nu = 1$). Based on our experimental observations, we derive fitting functions to estimate accuracy and BitOps for each decay value of $\nu$ on CIFAR10 and ImageNet datasets. We note that the behavior of these functions near zero differs from their behavior in other intervals. Specifically, for CIFAR10, both accuracy and BitOps can be accurately estimated using different quadratic polynomial functions in both intervals of $\nu$. In contrast, for ImageNet, while a quadratic polynomial remains effective near zero, a logarithmic function provides a better fit for accuracy when $\nu$ lies within the range of $10^{-6}$ to 1. For CIFAR10, the accuracy and BitOps as functions of $\nu$ are defined piecewise as:

$$\text{Accuracy}(\nu) = \begin{cases} 2.52 \times 10^{10}\nu^2 - 2.52 \times 10^7\nu + 95.95, & 0 \leq \nu \leq 10^{-3}, \\ 52.97\,\nu^2 - 70.10\,\nu + 91.55, & 10^{-3} < \nu \leq 1, \end{cases} \tag{16}$$

$$\text{BitOps}(\nu) = \begin{cases} 6.599 \times 10^{12}\nu^2 - 6.599 \times 10^9\nu + 9.926, & 0 \leq \nu \leq 10^{-3}, \\ 9.135\,\nu^2 - 10.863\,\nu + 1.805, & 10^{-3} < \nu \leq 1. \end{cases} \tag{17}$$

For ImageNet, the corresponding piecewise functions are given as:

$$\text{Accuracy}(\nu) = \begin{cases} 9.966 \times 10^{14}\nu^2 - 1.0 \times 10^9\nu + 75.1, & 0 \leq \nu \leq 10^{-6}, \\ -0.332\,(\log_{10}\nu)^2 - 5.780\,\log_{10}\nu + 50.1, & 10^{-6} < \nu \leq 1, \end{cases} \tag{18}$$

$$\text{BitOps}(\nu) = \begin{cases} 2.326 \times 10^{16}\nu^2 - 2.328 \times 10^{10}\nu + 31.49, & 0 \leq \nu \leq 10^{-6}, \\ 0.166\,(\log_{10}\nu)^2 + 0.285\,(\log_{10}\nu) + 0.146, & 10^{-6} < \nu \leq 1. \end{cases} \tag{19}$$

It is noteworthy that other research studies also have such hyperparameter that need to be adjusted to get the final results. For instance, ProxylessNAS (Cai et al. (2018)) uses an $\omega$ that controls the trade-off between accuracy and latency. Network Slimming (Liu et al. (2017b)) and others that prune via a penalty on batch-norm scales often try a few $\lambda$ values and pick the one that yields the desired sparsity. In Network Slimming (Liu et al. (2017b)), for CIFAR10 and SVHN datasets, the sparsity-controlling hyperparameter $\lambda$ (balancing empirical loss and sparsity) is selected via grid search over $\{10^{-3}, 10^{-4}, 10^{-5}\}$. For ImageNet, in case of VGG, it sets $\lambda = 10^{-4}$.

## A.5 Applicability of DNNs for Widely Used Applications

DNNs – particularly compact CNN-based models – remain widely used for inference in deployment settings because they offer a favorable balance of accuracy, latency, and energy. In latency- and energy-constrained applications (mobile, smart home devices, wearable devices, AR/VR, etc), simpler DNN architectures are preferred over extremely large models due to their lower computational and memory requirements, which translate to faster inference and lower power consumption. Numerous recent work across different domains also underscore the continued dominance of optimized DNNs in deployment (Khare et al. (2024); Zhang et al. (2024); Lin et al. (2022); Behnam et al. (2023); Sahni et al. (2021)).

## A.6 Potential Joint Differentiable NAS-Quantization Search for Transformers

Traditional DARTS-based NAS operates on CNN architectures –e.g., choosing conv vs. pooling operations, cell connectivity patterns, etc. – whereas Transformer-based NAS focuses on different macro-parameters and layer structures. In this architecture, critical design dimensions include the network depth (number of self-attention layers), the embedding dimension of tokens, and the number of attention heads per layer, among others. These factors have a large impact on model accuracy and efficiency. Recent NAS approaches aim to automate this. For instance, AutoFormer (Chen et al. (2021)) introduces a one-shot NAS Transformer, treating depth, embedded size, and heads as search parameters rather than fixed values. It uses weight-sharing across Transformer blocks to train many architecture variants simultaneously. This illustrates that Transformer-based NAS requires handling a structured, layer-wise search space (with choices like how many layers or heads), which is a different scenario than the cell-based micro-search of DARTS. Nonetheless, the differentiable search philosophy can be extended to the Transformer. For example, we can assign continuous relaxation variables to Transformer choices (e.g., a parameter that selects between smaller or larger embedding sizes), similar to how DARTS selects between candidate operations.

We can potentially formulate the joint NAS and quantization search as a *bi-level optimization* problem. Let $\alpha$ denote the architecture parameters (e.g., network depth, embedding dimension, number of heads, etc.), and let $\gamma$ denote the quantization parameters (e.g., bit-precision for model weights). We use $w$ notation for the network weights. The goal is to find the optimal architecture (including quantization policy) that minimizes validation loss $\mathcal{L}_{\text{val}}$, while the weights $w$ are trained to minimize training loss $\mathcal{L}_{\text{train}}$. This can be expressed as:

$$\min_{\alpha, \gamma} \quad \mathcal{L}_{\text{val}}\big(w^*(\alpha, \gamma)\, ;\; \alpha, \gamma\big)\,, \tag{20}$$

$$\text{s.t.} \quad w^*(\alpha, \gamma) \;=\; \arg\min_{w} \mathcal{L}_{\text{train}}(w\, ;\; \alpha, \gamma)\,. \tag{21}$$

Equation equation 20 is the *outer loop* (architecture/precision optimization) and equation 21 is the *inner loop* (weight training for a given architecture). Solving this bi-level problem directly is challenging because $\alpha$ and $\gamma$ are involved in the training process of $w$.

Alternatively, we can use the more optimized equations:

$$\min_{\alpha} \quad \mathcal{L}_{\text{val}}(\omega^*(\alpha), \gamma^*(\alpha), \alpha)$$
$$\text{s.t.} \quad \omega^*(\alpha), \gamma^*(\alpha) = \underset{\omega, \gamma}{\text{argmin}} \quad \mathcal{L}_{\text{train}}(\omega, \gamma, \alpha) \tag{22}$$

Following the $\nabla\texttt{QDARTS}$ approach, we relax the architecture search space to be continuous so that $\alpha$ and $\gamma$ can be optimized with gradient-based methods alongside $w$.

**Continuous Relaxation of the Search Space (Architecture & Quantization)**

Transformer-based architectures have several discrete parameters such as the number of Transformer layers $L$, the embedding (hidden) dimension $d$, the number of attention heads $h$, and the quantization precision $\gamma$. We incorporate these parameters into the search.

To make these choices differentiable, we represent each as a weighted softmax over possible options, parameterized by $\alpha$ or $\gamma$. This is analogous to the differentiable relaxation in $\nabla$QDARTS, but extended to non-operator choices and quantization. We detail each below.

**Depth ($L$):**  We let the search consider a set of possible layer counts $\{L_1, L_2, \ldots, L_m\}$ (e.g., $L \in \{2, 4, 6\}$ in the search space). We assign architecture parameters $\alpha_L^{(i)}$ for each candidate $L_i$. A softmax over these yields selection probabilities for each depth:

$$p_L^{(i)} = \frac{\exp(\alpha_L^{(i)})}{\sum_{j=1}^{m} \exp(\alpha_L^{(j)})}, \quad i = 1, \ldots, m. \tag{23}$$

We construct a *SuperNet* that can accommodate the maximum depth $L_{\max} = \max\{L_i\}$. During a forward pass, we can blend the outputs from different depths according to $p_L$. For example, let $z_\ell$ be the output of the network truncated to $\ell$ layers. We form the final output as a weighted sum of intermediate outputs: $z_{\text{final}} = \sum_{i=1}^{m} p_L^{(i)} z_{L_i}$. This effectively creates a continuous mixture over architectures of different depths. During the search, $\alpha_L$ is learned such that it increasingly favors the optimal depth. In the end, we select the discrete depth $L = L_i$ with the highest probability $p_L^{(i)}$.

**Embedding Dimension ($d$):**  We consider a set of possible hidden dimensions $\{d_1, d_2, \ldots, d_n\}$ (for instance $d \in \{128, 256, 512\}$). Similarly, we introduce architecture parameters $\alpha_d^{(j)}$ for each candidate $d_j$ and form a softmax:

$$p_d^{(j)} = \frac{\exp(\alpha_d^{(j)})}{\sum_{k=1}^{n} \exp(\alpha_d^{(k)})}, \quad j = 1, \ldots, n. \tag{24}$$

To implement a mixture of dimensions, one can build the model with the largest dimension $d_{\max}$ and use a masking or gating mechanism. For example, each linear layer in the Transformer can be designed to have $d_{\max}$ dimension, and smaller $d_j$ configurations are obtained by only using the first $d_j$ neurons (the remaining $d_{\max} - d_j$ neurons can be masked out or simply not activated). In the continuous relaxation, we can weight the contribution of each dimension corresponding to a given $d_j$ by $p_d^{(j)}$. Conceptually, this means the effective embedding dimension during a forward pass is an average of all candidate dimensions, weighted by $p_d$. The architecture gradients adjust $\alpha_d$ to favor the dimension that best balances accuracy and complexity. After searching, we discretize by choosing the dimension $d = d_j$ with the highest probability.

**Number of Heads ($h$).**  We also search over the number of attention heads. Let the candidate head counts be $\{h_1, h_2, \ldots, h_q\}$ (e.g., $h \in \{4, 8, 16\}$ heads). We assign parameters $\alpha_h^{(r)}$ for each $h_r$ and obtain:

$$p_h^{(r)} = \frac{\exp(\alpha_h^{(r)})}{\sum_{s=1}^{q} \exp(\alpha_h^{(s)})}, \quad r = 1, \ldots, q. \tag{25}$$

Including different head counts in one *SuperNet* is achievable by computing multi-head attention with a maximum number of heads $h_{\max}$, then treating some heads as inactive for smaller $h$. In practice, if $h_{\max}$ is the largest option, an attention operation with fewer heads (say $h_r < h_{\max}$) can be represented by grouping or masking heads in the $h_{\max}$-head attention. In the relaxed formulation, we can compute attention outputs for all $h_{\max}$ heads and then weight/select the appropriate combination corresponding to each candidate $h_r$.

The probabilities $p_h^{(r)}$ then mix these outcomes, and ultimately we pick the discrete $h$ that has the highest weight.

**Quantization Precision ($\gamma$).** For weight quantization, we consider a set of bit-widths (precisions) $\{\gamma_1, \gamma_2, \ldots, \gamma_t\}$. We introduce a set of quantization architecture parameters $\gamma^{(u)}$ (overloading notation $\gamma$ to also denote the set of softmax weights for precision choices) for each candidate bit-width $\gamma_u$. The softmax relaxation is:

$$p_\gamma^{(u)} = \frac{\exp(\gamma^{(u)})}{\sum_{v=1}^t \exp(\gamma^{(v)})}, \quad u = 1, \ldots, t. \tag{26}$$

To combine different precisions in one model, we leverage the idea of *mixed-precision SuperNet in $\nabla$`QDARTS`*. During a forward pass, we can simulate low-precision weights by quantizing the underlying high-precision weights $w$ to the required bit-width. Let $w_{\gamma_u}$ denote the network weights quantized to $\gamma_u$ bits. We then compute the network output for each precision candidate, $z_{\gamma_u} = f(x; w_{\gamma_u})$ The overall output is a weighted sum of these precision-specific outputs: $z_{\text{final}} = \sum_{u=1}^t p_\gamma^{(u)} z_{\gamma_u}$. This guides $\gamma$ towards the bit-width that best trades accuracy for compression. After search, we select the bit-width with the highest probability (or sometimes the smallest bit-width that maintains accuracy, depending on the target criteria).

Through the above continuous relaxation, the architecture parameters $\alpha$ and quantization parameters $\gamma$ now define a *soft architecture* that is differentiable. The effective choices of $L$, $d$, $h$, and bit-width $\gamma$ are implicitly given by the probability weights $p_L^{(i)}, p_d^{(j)}, p_h^{(r)}, p_\gamma^{(u)}$. Initially, these weights are typically uniform. During training (solving the bilevel problem), $\alpha$ and $\gamma$ are updated (in the outer loop) using gradients from the validation loss, while $w$ is optimized on the training loss (inner loop). Over time, the softmax weights concentrate around the best-performing architectural configuration. Once training converges, we obtain a discrete architecture by picking the argmax option for each hyperparameter (or by sampling according to $p$ and retraining, as done in $\nabla$`QDARTS`). This yields a Transformer architecture with a specific depth $L$, embedding dimension $d$, number of heads $h$, and weight precision $\gamma$ bits, all chosen to minimize validation loss.

### Complexity-Aware Loss Function

Simply optimizing for validation loss may lead to choosing the largest model and highest precision (to maximize accuracy). To steer the search towards efficient models, we incorporate a *complexity penalty* in the objective, as outlined in the original $\nabla$`QDARTS` methodology. We quantify the model's complexity in terms of estimated parameter count and computational cost, and add this to the validation loss with a tunable weight.

First, we estimate the number of parameters $N_{\text{params}}(L, d)$ for a Transformer with $L$ layers and embedding dimension $d$ (assuming the feed-forward network dimension is $f$, and keeping other aspects constant). A single Transformer layer consists of the multi-head self-attention and the feed-forward sub-layer. The attention part contributes roughly $4d^2$ parameters (for the query, key, value, and output projection matrices), and the feed-forward part contributes about $2df$ (for the two linear Transformations) (Chen et al. (2021); Vaswani et al. (2017)). Thus, the total parameter count scales as:

$$N_{\text{params}}(L, d) \approx L\left(4d^2 + 2df\right), \tag{27}$$

where $f$ is typically a multiple of $d$ (for example $f = 4d$ in many Transformer designs). This formula provides a rough but useful estimate of model size given $L$ and $d$.

Next, we define a complexity cost $\mathcal{C}(\alpha, \gamma)$ that incorporates both model size and compute cost, taking into account quantization. We have two sources of complexity to consider: (1) model memory/computation due to the number of parameters (which is reduced by lower precision), and (2) the runtime operations for attention and feed-forward computations per layer. We denote by $O_{\text{attn}}(d, h)$ the cost (e.g., number of operations) of the attention mechanism in a single layer with dimension $d$ and $h$ heads, and by $O_{\text{ffn}}(d, f)$ the cost of the feed-forward layer. We combine these as follows:

$$\mathcal{C}(\alpha,\gamma) \;=\; L\Big(4\,d^2 \;+\; 2\,d\,f\Big)\frac{\gamma}{32} \;+\; \eta\,L\Big(O_{\mathrm{attn}}(d,h) \;+\; O_{\mathrm{ffn}}(d,f)\Big), \tag{28}$$

In this expression, the first term $L(4d^2 + 2df)\frac{\gamma}{32}$ represents the effective model size scaled by the precision factor $\gamma/32$. Here $\gamma$ (in bits) is normalized by 32, since a full-precision model is 32-bit; Lower $\gamma$ directly reduces this term, favoring low-bit quantized models. The second term $\eta\,L\,(O_{\mathrm{attn}} + O_{\mathrm{ffn}})$ captures the estimated computation cost per layer (which could be measured in FLOPs or a proxy for latency). The coefficient $\eta$ is a weighting factor to balance the importance of operation count relative to parameter count; it can be set to reflect hardware characteristics (for instance, if memory is more constrained than compute, $\eta$ can be set smaller, and vice versa). Both components are multiplied by $L$ because a deeper network repeats the cost at each layer. Note that in the continuous relaxation setting, $L$, $d$, $h$, and $\gamma$ in equation equation 28 are the soft values (e.g., expected layer count, expected dimension, etc., under the current $p$ distribution).

Finally, the above complexity term is added to the validation loss to form the augmented objective for the outer optimization. The combined objective (to be minimized with respect to $\alpha,\gamma$) is:

$$\mathcal{L}_{\mathrm{val}}\big(w^*(\alpha,\gamma);\,\alpha,\gamma\big) \;+\; \lambda\,\mathcal{C}(\alpha,\gamma) \tag{29}$$

where $\lambda$ is a regularization hyperparameter that controls the trade-off between accuracy and complexity. A higher $\lambda$ places more emphasis on the complexity penalty, pushing the search towards architectures that are smaller or computationally cheaper (e.g., fewer layers, smaller $d$, fewer heads, lower precision), potentially at some loss in accuracy. A lower $\lambda$ makes the search focus more on accuracy (the validation loss term) with less regard for model size or speed. By tuning $\lambda$, we can obtain a family of architectures along the Pareto frontier of accuracy vs. efficiency.

It is noteworthy that the implementation and getting results for the Transformers are left for future work. Through the implementation phase and getting results, some of the equations might need some modifications.

