# OpenReview forum: "∇QDARTS: Quantization as an Elastic Dimension to Differentiable NAS"
_TMLR — Accepted by TMLR_

### Review · Reviewer_mDKj · 2025-02-25

**Summary Of Contributions:**

This paper proposes novel methods to include mixed-precision quantization into the differentiable NAS algorithm on the DARTS search space. The proposed framework enables a flexible navigation in accuracy-BitOps trade-off space by including a complexity regularizer into the search process. Highly efficient arhchitectures are discovered from the search, outperforming both previous NAS methods and manually-designed quantized models.

**Audience:**

Yes

**Broader Impact Concerns:**

There's no concern on the broader impact of this paper.

**Claims And Evidence:**

Yes

**Requested Changes:**

Please discuss the extension of the proposed method on transformer search space, such as the one proposed in AutoFormer: Searching Transformers for Visual Recognition. https://arxiv.org/abs/2107.00651

**Strengths And Weaknesses:**

## Strength

1. This paper is well-written and very easy to follow
2. Detailed discussion on related work is provided, making the proposed method well-motivated and well-positioned
3. Extensive experiments and ablation studies are conducted, including comparison with both NAS baselines and other non-NAS quantization methods.
4. Experimental results shows the proposed method is effective in searching for efficient mixed-precision architectures

## Weakness

The main weakness of the paper is that the exploration is solely based on the DARTS search space, which is old in today's standard and is limiting on the performance that it can achieve. The impact of this work can be further strengthened by showing the extension of the proposed training method on more advanced differentiable NAS search space, especially those for Transformers. One possible reference is the AutoFormer search space proposed in https://arxiv.org/abs/2107.00651

---

> ### Author Response · Authors · 2025-03-17
> **Response to Reviewer mDKj**
>
> ### **Applicability of DNNs for Widely Used Applications:**
> DNNs – particularly compact CNN-based models – remain widely used for inference in deployment settings because they offer a favorable balance of accuracy, latency and energy. In latency- and energy-constrained applications (mobile, smart home device, wearable devices, AR/VR, etc), simpler DNN architectures are preferred over extremely large models due to their lower computational and memory requirements, which translate to faster inference and lower power consumption. Numerous recent works across different domains underscore the continued dominance of optimized DNNs in deployment [RefAAA] [RefBBB] [RefCCC] [RefDDD] [RefEEE] [RefFFF] [RefGGG] [RefHHH] [RefIII] [RefGGG]
>
>
> Although transformer-based models are becoming popular, resource-efficient DNNs remain indispensable in deployment (especially on edge platforms). Accordingly, like all the extensive baselines, focusing our ∇QDARTS method on such DNN search spaces is thus well-justified, as it targets the prevalent use-case of optimizing neural networks for real-world constraints like latency, power, and memory.
>
> ### **Differences of NAS in CNN vs. Transformer Domains:**
> Traditional DARTS-based NAS operates on CNN architectures --e.g. choosing conv vs. pooling operations, cell connectivity patterns, etc. – whereas transformer-based NAS focuses on different macro-parameters and layer structures. In this architecture, critical design dimensions include the network depth (number of self-attention layers), the embedding dimension of tokens, and the number of attention heads per layer, among others. These factors have a large impact on model accuracy and efficiency. Recent NAS approaches aim to automate this. For instance, AutoFormer [RefKKK] introduces a one-shot NAS transformers, treating depth, embedded size, and heads as search parameters rather than fixed values. It uses weight-sharing across transformer blocks to train many architecture variants simultaneously. This illustrates that transformer-based NAS requires handling a structured, layer-wise search space (with choices like how many layers or heads), which is a different scenario than the cell-based micro-search of DARTS. Nonetheless, the differentiable search philosophy can be extended to the transformer. For example, we can assign continuous relaxation variables to transformer choices (e.g., a parameter that selects between smaller or larger embedding sizes), similar to how DARTS selects between candidate operations.

---

> ### Author Response · Authors · 2025-03-17
> **Response to Reviewer mDKj-Part2**
>
> ### **Joint Differentiable NAS-Quantization Search for Transformers**:
>
> We can potentially formulate the joint NAS and quantization search as a \emph{bi-level optimization} problem. Let $\alpha$ denote the architecture parameters (e.g., network depth, embedding dimension, number of heads, etc.), and let $\gamma$ denote the quantization parameters (e.g., bit-precision for model weights). We use $w$ notation for the network weights. The goal is to find the optimal architecture (including quantization policy) that minimizes validation loss $L_{\rm val}$
> , while the weights $w$ are trained to minimize training loss $\mathcal{L}_{\text{train}}$. This can be expressed as:
>
> $$
> \min_{\alpha,\gamma} L_{\rm val}\bigl(w^*(\alpha,\gamma);\alpha,\gamma\bigr)
> \quad \mbox{s.t.} \quad
> w^*(\alpha,\gamma) = \arg\min_{w} L_{\rm train}(w;\alpha,\gamma)
> $$
>
>
> First equation is the outer loop (architecture/precision optimization) and second one is the inner loop (weight training for a given architecture). Solving this bi-level problem directly is challenging because $\alpha$ and $\gamma$ are involved in the training process of $w$.
>
>
> Alternatively, we can use the more optimized equation:
>
> $$
> \min_{\alpha}  L_{\rm val}\bigl(\omega^*(\alpha), \gamma^*(\alpha), \alpha\bigr)
> \quad \mbox{s.t.} \quad
> \omega^*(\alpha), \gamma^*(\alpha) = \underset{\omega,\gamma}{\mathrm{argmin}}L_{\rm train}(\omega, \gamma, \alpha)
> $$
>
>
>
> Following the ∇QDARTS approach, we relax the architecture search space to be continuous so that $\alpha$ and $\gamma$ can be optimized with gradient-based methods alongside $w$.
>
> #### **Continuous Relaxation of the Search Space (Architecture \& Quantization):**
>
> Transformer-based architectures have several discrete parameters such as the number of transformer layers $L$, the embedding (hidden) dimension $d$, the number of attention heads $h$, and the quantization precision $\gamma$. We incorporate these parameters into the search.
>
> To make these choices differentiable, we represent each as a weighted softmax over possible options, parameterized by $\alpha$ or $\gamma$. This is analogous to the differentiable relaxation in ∇QDARTS, but extended to non-operator choices and quantization. We detail each below.
>
> **Depth ($L$):** We let the search consider a set of possible layer counts $\{L_1, L_2, \dots, L_{m}\}$ (e.g., $L\in\{2,4,6\}$ in the search space). We assign architecture parameters $\alpha_L^{(i)}$ for each candidate $L_i$. A softmax over these yields selection probabilities for each depth:
>
> \begin{equation}
> p_L^{(i)} = \frac{\exp(\alpha_L^{(i)})}{\sum_{j=1}^{m}\exp(\alpha_L^{(j)})}, \quad i=1,\dots,m.
> \end{equation}
>
> We construct a \supernet that can accommodate the maximum depth $L_{\max} = \max\{L_i\}$. During a forward pass, we can blend the outputs from different depths according to $p_L$. For example, let $z_{\ell}$ be the output of the network truncated to $\ell$ layers. We form the final output as a weighted sum of intermediate outputs: $z_{\text{final}} = \sum_{i=1}^{m} p_L^{(i)}\, z_{L_i}$. This effectively creates a continuous mixture over architectures of different depth. During the search, $\alpha_L$ is learned such that it increasingly favors the optimal depth. In the end, we select the discrete depth $L = L_i$ with the highest probability $p_L^{(i)}$.
>
> **Embedding Dimension ($d$):** We consider a set of possible hidden dimensions $\{d_1, d_2, \dots, d_{n}\}$ (for instance $d\in\{128, 256, 512\}$). Similarly, we introduce architecture parameters $\alpha_d^{(j)}$ for each candidate $d_j$ and form a softmax:
>
> \begin{equation}
> p_d^{(j)} = \frac{\exp(\alpha_d^{(j)})}{\sum_{k=1}^{n}\exp(\alpha_d^{(k)})}, \quad j=1,\dots,n.
> \end{equation}
>
> To implement a mixture of dimensions, one can build the model with the largest dimension $d_{\max}$ and use a masking or gating mechanism. For example, each linear layer in the transformer can be designed to have $d_{\max}$ dimension, and smaller $d_j$ configurations are obtained by only using the first $d_j$ neurons (the remaining $d_{\max}-d_j$ neurons can be masked out or simply not activated). In the continuous relaxation, we can weigh the contribution of each dimension corresponding to a given $d_j$ by $p_d^{(j)}$. Conceptually, this means the effective embedding dimension during a forward pass is an average of all candidate dimensions, weighted by $p_d$. The architecture gradients adjust $\alpha_d$ to favor the dimension that best balances accuracy and complexity. After searching, we discretize by choosing the dimension $d = d_j$ with the highest probability.

---

> ### Author Response · Authors · 2025-03-17
> **Response to Reviewer mDKj-Part3**
>
> **Number of Heads ($h$).** We also search over the number of attention heads. Let the candidate head counts be $\{h_1, h_2, \dots, h_{q}\}$ (e.g., $h\in\{4,8,16\}$ heads). We assign parameters $\alpha_h^{(r)}$ for each $h_r$ and obtain:
>
> \begin{equation}
> p_h^{(r)} = \frac{\exp(\alpha_h^{(r)})}{\sum_{s=1}^{q}\exp(\alpha_h^{(s)})}, \quad r=1,\dots,q.
> \end{equation}
>
> Including different head counts in one \supernet is achievable by computing multi-head attention with a maximum number of heads $h_{\max}$, then treating some heads as inactive for smaller $h$. In practice, if $h_{\max}$ is the largest option, an attention operation with fewer heads (say $h_r < h_{\max}$) can be represented by grouping or masking heads in the $h_{\max}$-head attention.
> In the relaxed formulation, we can compute attention outputs for all $h_{\max}$ heads and then weight/select the appropriate combination corresponding to each candidate $h_r$. The probabilities $p_h^{(r)}$ then mix these outcomes, ultimately we pick the discrete $h$ that has the highest weight.
>
>
>
> **Quantization Precision ($\gamma$).** For weight quantization, we consider a set of bit-widths (precisions) $\{\gamma_1, \gamma_2, \dots, \gamma_{t}\}$. We introduce a set of quantization architecture parameters $\gamma^{(u)}$ (overloading notation $\gamma$ to also denote the set of softmax weights for precision choices) for each candidate bit-width $\gamma_u$. The softmax relaxation is:
> \begin{equation}
> p_\gamma^{(u)} = \frac{\exp(\gamma^{(u)})}{\sum_{v=1}^{t}\exp(\gamma^{(v)})}, \quad u=1,\dots,t.
> \end{equation}
>
> To combine different precisions in one model, we leverage the idea of \emph{mixed-precision \supernet in ∇QDARTS}. During a forward pass, we can simulate low-precision weights by quantizing the underlying high-precision weights $w$ to the required bit-width. Let $w_{\gamma_u}$ denote the network weights quantized to $\gamma_u$ bits.
> We then compute the network output for each precision candidate, $z_{\gamma_u} = f(x; w_{\gamma_u})$
> The overall output is a weighted sum of these precision-specific outputs: $z_{\text{final}} = \sum_{u=1}^{t} p_\gamma^{(u)} z_{\gamma_u}$.
>
> This guides $\gamma$ towards the bit-width that best trades accuracy for compression. After search, we select the bit-width with highest probability (or sometimes the smallest bit-width that maintains accuracy, depending on the target criteria).
>
>
> Through the above continuous relaxation, the architecture parameters $\alpha$ and quantization parameters $\gamma$ now define a \textit{soft architecture} that is differentiable. The effective choices of $L$, $d$, $h$, and bit-width $\gamma$ are implicitly given by the probability weights $p_L^{(i)}, p_d^{(j)}, p_h^{(r)}, p_\gamma^{(u)}$. Initially, these weights are typically uniform. During training (solving the bilevel problem), $\alpha$ and $\gamma$ are updated (in the outer loop) using gradients from the validation loss, while $w$ is optimized on the training loss (inner loop). Over time, the softmax weights concentrate around the best-performing architectural configuration. Once training converges, we obtain a discrete architecture by picking the argmax option for each hyperparameter (or by sampling according to $p$ and retraining, as done in ∇QDARTS). This yields a transformer architecture with a specific depth $L$, embedding dimension $d$, number of heads $h$, and weight precision $\gamma$ bits, all chosen to minimize validation loss.
>
> #### **Complexity-Aware Loss Function:**
>
> Simply optimizing for validation loss may lead to choosing the largest model and highest precision (to maximize accuracy). To steer the search towards efficient models, we incorporate a \emph{complexity penalty} in the objective, as outlined in the original ∇QDARTS methodology. We quantify the model's complexity in terms of estimated parameter count and computational cost, and add this to the validation loss with a tunable weight.
>
> First, we estimate the number of parameters $N_{\text{params}}(L,d)$ for a transformer with $L$ layers and embedding dimension $d$ (assuming the feed-forward network dimension is $f$, and keeping other aspects constant). A single transformer layer consists of the multi-head self-attention and the feed-forward sub-layer. The attention part contributes roughly $4d^2$ parameters (for the query, key, value, and output projection matrices), and the feed-forward part contributes about $2d\,f$ (for the two linear transformations) [RefKKK][RefLLL].
> Thus, the total parameter count scales as:
>
> $$
> N_{\rm params}(L,d) \approx L\Bigl(4d^2 + 2df\Bigr)
> $$
>
> where $f$ is typically a multiple of $d$ (for example $f=4d$ in many Transformer designs). This formula provides a rough but useful estimate of model size given $L$ and $d$.

---

> ### Author Response · Authors · 2025-03-17
> **Response to Reviewer mDKj-Part4**
>
> Next, we define a complexity cost $\mathcal{C}(\alpha,\gamma)$ that incorporates both model size and compute cost, taking into account quantization. We have two sources of complexity to consider: (1) model memory/computation due to the number of parameters (which is reduced by lower precision), and (2) the runtime operations for attention and feed-forward computations per layer. We denote by $O_{\text{attn}}(d,h)$ the cost (e.g., number of operations) of the attention mechanism in a single layer with dimension $d$ and $h$ heads, and by $O_{\text{ffn}}(d,f)$ the cost of the feed-forward layer. We combine these as follows:
>
> $$
> C(\alpha,\gamma) = L\Bigl(4d^2 + 2df\Bigr)\frac{\gamma}{32} + \eta L\Bigl(O_{\rm attn}(d,h) + O_{\rm ffn}(d,f)\Bigr)
> $$
>
>
> In this expression, the first term $L(4d^2+2df)\,\frac{\gamma}{32}$ represents the effective model size  scaled by the precision factor $\gamma/32$. Here $\gamma$ (in bits) is normalized by 32, since a full-precision model is 32-bit;
>
> Lower $\gamma$ directly reduces this term, favoring low-bit quantized models. The second term $\eta\,L\,(O_{\text{attn}} + O_{\text{ffn}})$ captures the estimated computation cost per layer (which could be measured in FLOPs or a proxy for latency). The coefficient $\eta$ is a weighting factor to balance the importance of operation count relative to parameter count; it can be set to reflect hardware characteristics (for instance, if memory is more constrained than compute, $\eta$ can be set smaller, and vice versa). Both components are multiplied by $L$ because a deeper network repeats the cost each layer. Note that in the continuous relaxation setting, $L$, $d$, $h$, and $\gamma$ in the above equation are soft values (e.g., expected layer count, expected dimension, etc., under the current $p$ distribution).
>
>
> Finally, the above complexity term is added to the validation loss to form the augmented objective for the outer optimization. The combined objective (to be minimized with respect to $\alpha,\gamma$) is:
>
> $$
> L_{\rm val}\bigl(w^*(\alpha,\gamma); \alpha,\gamma\bigr) + \lambda C(\alpha,\gamma)
> $$
>
>
> where $\lambda$ is a regularization hyperparameter that controls the trade-off between accuracy and complexity. A higher $\lambda$ places more emphasis on the complexity penalty, pushing the search towards architectures that are smaller or computationally cheaper (e.g. fewer layers, smaller $d$, fewer heads, lower precision), potentially at some loss in accuracy. A lower $\lambda$ makes the search focus more on accuracy (the validation loss term) with less regard for model size or speed. By tuning $\lambda$, we can obtain a family of architectures along the Pareto frontier of accuracy vs. efficiency.
>
> **It is noteworthy that the implementation and getting results for the transformers are left for future work. Through the implementation phase and getting initial results, some of the equations might need some modifications.**
>
> **Please refer to the revised version for better formatted response.
>
> **References:**
>
> [RefAAA]: SuperFedNAS: Cost-Efficient Federated Neural Architecture Search for On-Device Inference, ECCV, 2024.
>
> [RefBBB]: CamoNet: On-Device Neural Network Adaptation With Zero Interaction and Unlabeled Data for Diverse Edge Environments, IEEE Transactions on Mobile Computing, 2024.
>
> [RefCCC]: ElasticDNN: On-Device Neural Network Remodeling for Adapting Evolving Vision Domains at Edge, IEEE Transactions on Computers, 2024.
>
> [RefDDD] On-Device Training Under 256KB Memory, NeurIPS, 2022.
>
> [RefEEE] Towards On-device Learning on the Edge: Ways to Select Neurons to Update under a Budget Constraint, WACV 2024.
>
> [RefFFF] nn-Meter: Towards Accurate Latency Prediction of Deep-Learning Model Inference on Diverse Edge Devices, MobiSys 2021.
>
> [RefGGG]: MCUNetV2: Memory-Efficient Patch-based Inference for Tiny Deep Learning, NeurIPS 2021.
>
> [RefHHH]: Mediator: Characterizing and Optimizing Multi-DNN Inference for Energy Efficient Edge Intelligence, IISWC 2024.
>
> [RefIII]: Hardware-Software Co-design for Real-time Latency-accuracy Navigation in TinyML Applications, IEEE Micro, 2023.
>
>  [RefJJJ]: Ultra Low Complexity Deep Learning Based Noise Suppression, ICASSP, 2024.
>
> [RefKKK]: Autoformer: Searching transformers for visual recognition, IEEE/CVF international conference on computer vision, 2021
>
> [RefLLL]: Attention is all you need, Advances in Neural Information Processing Systems, 2017

---

### Review · Reviewer_gXNN · 2025-03-03

**Summary Of Contributions:**

The paper proposes a framework that integrates architecture and weight search with a mixed-precision quantization policy for both weights and activations. The proposal is evaluated experimentally.

**Audience:**

Yes

**Claims And Evidence:**

No

**Requested Changes:**

1. It is not clear why applying quantization is not straightforward in a NAS setting. We could (1) use the same bitwidth across all the layers of a model and then (2) do any standard NAS method. Please explain what are the issues of doing this.

2. "PTQ takes less time since it assumes that an expensive fully trained network is available to be quantized with fixed-bit and no extra fine-tuning is required". is this possible to be done for QDARTS? If yes, pleas present the results. If not, please explain with arguments why this is not possible.

3. Some of the works mentioned in section 2.3 are not included in the experimental section (for example DNAS). Moreover ,the reader will be able to understand better the SOTA compariosn if the works are grouped in the tables and maybe sort them based on accurcy or cost.

4. Please add the accuracy-BitOps trade-off plot for the proposal and on top of that include the SOTA works in this plot.

Minor Comments:

Table 3 is not referred in section 4.5. Please make sure that the tables and the figures are properly referred in the manuscript.

**Strengths And Weaknesses:**

Strengths:

1. The sections of the paper are well organized.

2. The paper presents a new method that combines differentiable NAS with mixed-precision search for both weight and activation.

3. The method is experimentally evalauted.

Weaknesses:

1. Some parts of the paper are not well written and it is difficult to follow.

2. The paper needs some clarifications in order to be techincally sound.

---

> ### Author Response · Authors · 2025-03-17
> **Response to Reviewer gXNN**
>
> ### **Question 1:**
>
> Applying quantization scheme across all layers in a standard NAS method is non-trivial due to multiple challenges as discussed in Section 1 and Section 2.3.
>
>
> First, a standard NAS method without quantization consideration does not ensure optimal performance in a quantized setting, since quantization efficiency depends on the neural architecture itself. As noted in Section 1, MobileNetV2 outperforms ResNet18 in full precision, but with 2-bit precision, ResNet18’s accuracy surpasses MobileNetV2’s [RefAA].
> This example demonstrates that different architectures respond differently to quantization.
> B incorporation quantization, the search mechanism may choose conv $5\times5$ instead of conv $3\times3$. Hence, finding architecture/weight first and then applying quantization is not an efficient solution.
>
> Second, NAS search space explosion makes it difficult to incorporate quantization during the joint architecture and weight search. For instance, weight sharing methods are proposed to alleviate the search space problem in NAS without quantization [RefAB][RefAC][RefAD]. Adding quantization makes it more challenging since now we need to search precision for each operation [RefAE][RefAB].
>
> Third, in order to get the lowest BitOps with the highest accuracy, we should employ mixed-precision and not fixed-precision. That needs to be applied to both weights and activations. For instance, if one applies a fixed 8-bit, it may get high accuracy at the cost of extra BitOpt. If one applies a fixed 2-bit, it may get low BitOps but at the cost of low accuracy. So, getting optimal trade-off points (the highest accuracy at the same BitOps or the lowest BitOps at the same accuracy) is not trivial.
>
> Figure 2 and Tables 2 and 3 show how different schemes affect the accuracy, BitOps and Memory footprint. ∇QDARTS Tackles the problems in a novel manner and reaches a better accuracy-BitOps trade-off.
>
>
> ### **Question 2:**
>
> Post-Training Quantization (PTQ) is indeed computationally efficient, but it is suboptimal in terms of accuracy as shown in Tables 2 and 3 since it does not search for quantization.
> ∇QDARTS performs quantization-aware architecture search jointly, rather than applying post quantization. As shown in our results, a naive PTQ approach leads to significant accuracy drops for DARTS-based architectures at lower bit-widths. For example, applying PTQ to a PC-DARTS architecture can result in drastic accuracy degradation (e.g., only about 1\% accuracy at 4-bit precision on ImageNet). This severe loss under ultra-low-bit quantization underscores why PTQ alone is not a viable solution.
> ∇QDARTS inevitably requires more search/training time than PTQ (since PTQ starts with a pre-trained network and does not search for proper bit-width for each layer), but this extra effort enables significantly better accuracy in mixed-precision settings. In summary, it is not feasible to simply apply PTQ in our NAS context without sacrificing accuracy, which is why ∇QDARTS co-designs the architecture and quantization during search.
>
> ### **Question 3:**
>
> We performed extensive study to show the capability of the proposed approach. As we mentioned in Section 4.2, there is no direct SOTA work that is comparable to ∇QDARTS.
> Our primary goal was to compare ∇QDARTS with approaches that jointly perform NAS and mixed-precision quantization, as these represent the most relevant benchmarks, enabling a more direct and fair comparison. DNAS, for instance, focuses on a fixed architecture (such as ResNet) and then searches for a per-layer mixed-precision configuration. This is a different problem setting from ∇QDARTS, which searches for the architecture and quantization policy together; hence, DNAS was not directly compared in our experiments.
>
>
> Hence, we defined three different baselines and compared ∇QDARTS with them. Moreover, we tried to find the closest work to ∇QDARTS (i.e., APQ , OQA, SPOS, and MNAS). We also compared with three non-NAS quantization methods, each with various precision settings. Totally, we compared against more than 20 baselines/settings.
>
>
> Moreover, categorizing these works is challenging, as their accuracy and complexity depend on various factors, including search space, dataset (e.g., DNAS only reports results on CIFAR-10), quantization levels, weight versus activation quantization, number of training epochs, employed FLOPS, etc. To clearly illustrate how these works have evolved over time, we have included the publication year in Table 1, providing readers with better context regarding their chronological development.

---

> ### Author Response · Authors · 2025-03-17
> **Response to Reviewer gXNN-Part2**
>
> ### **Question 4: **
> Figure https://imgur.com/a/OobBl23 illustrates the accuracy-BitOps trade-off for different versions of ∇QDARTS compared to other state-of-the-art methods. As shown, ∇QDARTS effectively pushes the frontier toward the top-left, achieving higher accuracy with fewer BitOps.
>
> ### **Minor Comments:**
> We mentioned Table 3 in Section 4.5.
>
> **Please refer to the revised version for better formatted response.
>
>
> **References:**
>
> [RefAA]: Once quantization-aware training: High performance extremely low-bit architecture search, CVPR 2021.
>
> [RefAB]: Soft weight-sharing for neural network compression, ICLR, 2017
>
> [RefAC]: Once-for-all: Train one network and specialize it for efficient deployment, ICLR2020
>
> [RefAD]:  Progressive neural architecture search, ECCV2018
>
> [RefAE]: Haq: Hardware-aware automated quantization with mixed precision, CVPR2019
>
> [RefAF]: Apq: Joint search for network architecture, pruning and quantization policy, CVPR2020

---

### Review · Reviewer_g4pC · 2025-03-04

**Summary Of Contributions:**

(***NOTE***: *in my review, I will use the text "QDARTS"  to denote the approach of this paper, i.e. without using the gradient symbol in the acronym.*)

1. This paper introduces QDARTS, an extension to the Differentiable Architecture Search (DARTS) framework that incorporates mixed-precision quantization as an additional search dimension. While DARTS and its variants (like PC-DARTS) have proven effective for neural architecture search through gradient-based optimization in a continuous domain, they operate in full precision and require separate quantization steps afterward, leading to suboptimal results. The key innovation in QDARTS is the seamless integration of mixed-precision search for both weights and activations directly into the differentiable architecture search process.

2. The authors formulate this as an extension of the bi-level optimization used in DARTS, where architecture parameters remain upper-level variables while weight and bit-precision parameters are jointly optimized as lower-level variables.

3. Experiments on CIFAR10 and ImageNet show that QDARTS can discover architectures that achieve high reductions in bit operations (up to 160x on CIFAR10 with 2-4 bit precision) while maintaining accuracy within 1-1.57% of full-precision DARTS models.

4. The method outperforms other quantization-aware NAS approaches like APQ, SPOS, OQA, and MNAS in terms of accuracy-efficiency trade-offs and search efficiency.

**Audience:**

Yes

**Broader Impact Concerns:**

The paper does not include a broader impact statement, which would be valuable given the potential applications of efficient neural networks.

**Claims And Evidence:**

Yes

**Requested Changes:**

# Recommended changes:
1. Evaluate QDARTS on at least one additional computer vision task beyond classification, such as object detection or semantic segmentation, to demonstrate its versatility across different vision tasks.
2. Include measurements of actual inference latency and energy consumption on real hardware platforms, particularly edge devices. This would provide more direct evidence of the practical benefits of the reduced computational requirements.
3. I am not sure if this is possible but it would be interesting to see ablation studies on different components of the bi-level optimization approach, such as comparing against a tri-level optimization where bit-precision is a separate level from weights or against a one-level approach where all parameters are optimized together.
4. Visualize and analyze the distribution of bit-precision selections across different layers and operations in the discovered architectures. While Figure 6 shows the cell structures, a more detailed analysis of which operations tend to use which precision would be insightful.
5. Include clearer guidelines for setting the complexity decay parameter ($\nu$) based on dataset characteristics, model requirements, or other factors. This is crucial for practitioners who want to apply QDARTS to new problems.

**Strengths And Weaknesses:**

# Strengths
1. **Novel extension of the DARTS Framework**: The paper extends DARTS to include quantization as a differentiable search dimension. This is a non-trivial extension that requires careful reformulation of the objective function and search space to accommodate bit-precision parameters.
    * The authors cleverly formulate bit-precision selection as a differentiable process by applying continuous relaxation similar to how DARTS handles architectural operations.

2. **Single-Shot End-to-End Framework**: Unlike multi-stage approaches like APQ or OQA that require pre-training or proxy networks, QDARTS performs joint architecture-weight-precision optimization in a single shot, which is useful in resource-constrained settings.

3. The complexity-aware loss function is interesting. The hyperparameter $\nu$ allows for flexible navigation of the accuracy-complexity trade-off space without requiring grid search.

4. **Comprehensive Experimental Validation**: The paper includes extensive comparisons against various baselines, including post-training quantization (PTQ), fixed-bit quantization-aware training (FBQAT), and SOTA methods. QDARTS has consistent advantages across different datasets and metrics.
    * The search and training efficiency is also better, achieving 3.1x and 1.54x improvements over APQ and OQA respectively.


# Weaknesses
1. **Sensitivity to Complexity Decay Parameter**: As shown in Table 7, the complexity decay parameter ($\nu$) has a substantial impact on the results, particularly for ImageNet (6.81% accuracy variation). The paper does not provide clear guidelines for selecting this parameter for new datasets.
2. **Limited Architecture Search Space**: The paper builds upon the DARTS search space, which primarily focuses on cell-based architectures. Exploring how QDARTS performs with other search spaces (e.g., MobileNet-like search spaces) would strengthen the generalizability claims.
3. **Task Diversity**: The evaluation is limited to image classification tasks. Testing on other computer vision tasks (object detection, segmentation) or even other domains would better demonstrate the versatility of the approach.

---

> ### Author Response · Authors · 2025-03-17
> **Response to Reviewer g4pC**
>
> ### **Question 1:**
> We respectfully emphasize that evaluating ∇QDARTS on tasks beyond image classification (such as object detection or semantic segmentation) is beyond the current scope of this manuscript. Our primary focus was to introduce and extensively evaluate ∇QDARTS in the context of differentiable NAS coupled with quantization, clearly establishing its strengths and efficacy within the classification setting. Furthermore, it is important to note that none of the existing baselines mentioned in our paper (e.g, Tables 1-5) have been evaluated or reported results on tasks other than classification.
> Additionally, in our response to _reviewer mDKj_ feedback, we discussed adapting ∇QDARTS to transformer-based architectures, which is used widely in object-detection and semantic segmentation[RefU, RefV, RefW].
> Nonetheless, the detailed extending our method, implementing, and performing experiments on other tasks would require significant additional implementation and exploration, which requires a separate, dedicated study.
> As a result, it has been left as a promising future direction.
>
> ### **Question 2:**
>
>  To the best of our knowledge, in the commercial market, there are currently no mainstream processors or NPUs that expose 2-bit ALUs to end-users, which is needed for us to deploy on real hardware. Hence, extracting results by deploying on real ASIC hardware is not feasible.
>  Most of the existing studies that study quantization, including those mentioned in papers, implement their methods using ML frameworks (e.g., PyTorch) and show efficiency of their methods with simulation or proxy metrics (e.g., latency, size, FLOPs) rather than demonstrating actual inference on real hardware.
>  Nevertheless, we employed two approaches to show how ∇QDARTS can impact latency and energy on hardware.
>
>
>
> #### **First Approach: Simulation \& Analysis**
>
> For the simulation-based results, we employed  Timeloop (i.e, DNN accelerator mapper/simulator) [RefS] and Accelergy (i.e., energy estimator) Ref[T] to evaluate latency/energy of the ∇QDARTS networks and compare them with the  PC-DARTS baseline in fp32 precision.
>
> In our analysis, we use Google’s Coral Edge TPU– a mobile edge AI accelerator with publicly available specifications [RefB][RefC][RefG]. The Edge TPU was chosen for its representative mobile-edge performance and its well-documented specifications, making it suitable to configure Timeloop/Accelergy tools.
>
>
> ##### **Simulation Configuration:**
> We set the clock frequency to 500 MHz and configure the Processing Elements (PE) array to match the Edge TPU’s compute capabilities. We assume  64 $\times$64 MAC units (4096 MACs total). At 500 MHz, 4096 MACs can perform almost 4 TOP/s.
> We set a scratchpad memory to 8 MB and the data width to 1 byte per element (since it stores 8-bit data primarily). Reading data from an 8MB on-chip SRAM costs around 8 pJ per 8-bit word [RefH][RefI][RefJ]. Since model footprints exceed 8MB, we include a DRAM level for main memory. We assume 320 pJ per 8-bit access for DRAM[RefH][RefI][RefJ]. We ensure the on-chip SRAM and PEs have sufficient bandwidth. E.g., the 8MB may be divided into banks.
>
> The granularity of the precision is per layer as mentioned in Figure 6 and we already know which operations work at what precision.
> We also use 0.5 pJ, 0.125 pJ, 0.03 pJ, 2 pJ and 8 pJ for MAC-8bit, MAC-4bit, MAC-2bit, MAC-16bit, and
> MAC-32bit, respectively [RefL][RefM].
>
> #### **Second Approach: FPGA Implementation**
>
> For FPGA implementation, we use AMD Xilinx Alveo U250 FPGA [RefO] that supports AMD-Xilinx’s FINN framework for low-bit neural networks via LUT-based logic for binary/ternary ops [RefN].
> To keep comparisons fair with simulation, we assume the same technology node and frequency and similar memory hierarchy constraints as the Timeloop/Accelergy study.
>
> ##### **FPGA Configuration:** FINN [RefN] yields a pipelined design where each layer processes data in a streaming fashion.
> The FINN accelerator would be structured to read weights from DDR, buffer them on chip, and stream the input through the pipeline. Each weight is reused for all necessary MACs (e.g. in convolutions) before loading new weights, minimizing redundant transfers.
>
> Because FINN allows increasing parallelism when quantization reduces resource usage, the low-precision models can be significantly unrolled, reducing cycles. In contrast, the FP32 model demands more resources per operation, forcing more pipeline stages, and higher cycle count, demanding more weights/feature maps stream from off-chip DDR4. Power is estimated based on device utilization and toggling activity.
> Lower-bit operations switch fewer transistors per compute. Quantized circuits consume less dynamic power per operation [RefP]. FINN’s post-implementation reports can provide an estimate of dynamic power given the switching rates for LUTs, DSPs, and clocks. Then, energy is computed as  $Power \times Latency$ [RefP][RefQ][RefR].

---

> ### Author Response · Authors · 2025-03-17
> **Response to Reviewer g4pC-Part2**
>
> #### **Comparing FPGA and Timeloop/Accelergy (ASIC) Results:**
>
> The following table summarizes the latency, power, and energy results for the three DNN models on the FPGA (FINN) and the ASIC (Timeloop/Accelergy). All results are for a single inference (batch=1).
>
> **Table  Estimated Latency and Energy Inference for Edge FPGA and ASIC under ∇QDARTS**
>
> | Model (Gbit-ops, Memory)                        | Platform | Latency (ms) | Compute Energy (mJ) | Memory Energy (mJ) | Total Energy (mJ) |
> |-------------------------------------------------|----------|--------------|---------------------|--------------------|-------------------|
> | FP32 Model (BitOps(G) = 536.6, Acc 76.1%)       | FPGA     | 60.7           | 8.1                   | 28.3                 | 36.4                |
> |                                                 | ASIC     | 30.7           | 2.2                   | 10.7                 | 12.9                |
> | Mixed 2/4/8-bit (BitOps(G) = 31.5m, Acc 75.1%)  | FPGA     | 15.6           | 1.3                   | 5.1                  | 6.4                 |
> |                                                 | ASIC     | 10.1           | 0.5                 | 2.0                | 2.5               |
> | Mixed 2/4-bit (BitOps(G) = 5.42, Acc 71.7%)     | FPGA     | 8.4            | 0.2                 | 1.8                | 2.0               |
> |                                                 | ASIC     | 5.2            | 0.15                | 0.6                | 0.75              |
>
> The FP32 design uses large bit-widths and drives off-chip DDR– approaching a high power draw– whereas the quantized designs use fewer bits, fewer bit toggles and skip most off-chip memory, leading to lower power and latency.  The mixed-precision models are far less demanding. The smallest 2/4-bit network executes in only a few milliseconds, with the ASIC drawing only a few hundred milliwatts (energy 0.75mJ). This aligns with expectations that ∇QDARTS quantization shrinks the workload by an order of magnitude, yielding lower total latency and energy versus the FP32 baseline.
>
> Please note the degree of ∇QDARTS improvement can vary based on the specific hardware device selected and its associated configuration parameters. Nevertheless, adopting ∇QDARTS consistently provides substantial advantages in terms of reduced energy consumption and lower latency, highlighting their potential for impact across various deployment scenarios.
>
>
> ### **Question 3:**
> We believe Section 4.8.1 and Figure 3 provide the requested information to the extent possible. Here, we study and compare different forms of bi-level optimization against ∇QDARTS under the CIFAR10 dataset. In addition, we show how ∇QDARTS outperforms one-level and tri-level optimizations.
>
> The One-level method requires only one forward pass per mini-batch per epoch, leading to the lowest search time but also the lowest accuracy and almost the same BitOps compared to ∇QDARTS. Compared to Tri-Level and Bi-Level-Other, ∇QDARTS reduces search time by 41.8% and 6.6%, respectively.
> ∇QDARTS reduces BitOps by 10% compared to Bi-Level-Other. It also provides 0.6\% higher accuracy compared to Tri-Level.
> It is important to note that larger datasets (e.g., ImageNet) are most likely to provide greater improvements; for instance, for search time, Tri-Level optimization times out after five days on ImageNet.
>
> ### **Question 4:**
>
> Figures in  https://imgur.com/a/Jkh3IMP show detailed analysis of which operations tend to use which precision for the discovered architecture in Figure 6 for both weight and activation, respectively. In the label of each bar, the first number shows the cell number and the second number shows the layer number.
>
> As shown, a significant portion of layers (about 54%) employ 2-bit weights, indicating heavy use of ultra-low precision weights in the network. Roughly, 32% of layers use 4-bit weights, and 14% use 8-bit weights.
> For the activation, we observe the same trends. The contributions of 2,4, and 8 bits are 52%, 26%, and 22%. Usually 2-bit precision is chosen for bigger kernels and 8-bit precision is selected for skip connections.
> It is noteworthy that the other layers in cells 2 and 5 are pooling layers, that we don't quantize them.

---

> ### Author Response · Authors · 2025-03-17
> **Response to Reviewer g4pC-Part3**
>
> ### **Question 5:**
> For all datasets, by adjusting the value of complexity decay parameter ($\nu$), we can either achieve the highest accuracy (at $\nu=0$) or minimize BitOps (at $\nu=1$). Based on our experimental observations, we derive fitting functions to estimate accuracy and BitOps for each decay value of $\nu$ on CIFAR10 and ImageNet datasets. We note that the behavior of these functions near zero differs from their behavior in other intervals. Specifically, for CIFAR10, both accuracy and BitOps can be accurately estimated using different quadratic polynomial functions in both intervals of $\nu$. In contrast, for ImageNet, while a quadratic polynomial remains effective near zero, a logarithmic function provides a better fit for accuracy when $\nu$ lies within the range of $10^{-6}$ to $1$.
>
> \begin{equation}
> \text{Accuracy}(\nu)=\begin{cases}
> 2.52 \times 10^{10} \nu^{2} - 2.52 \times 10^{7} \nu + 95.95, & 0 \leq \nu \leq 10^{-3}, \\
> 52.97\,\nu^{2} - 70.10\,\nu + 91.55, & 10^{-3} < \nu \leq 1,
> \end{cases}
> \end{equation}
>
> \begin{equation}
> \text{BitOps}(\nu)=\begin{cases}
> 6.599 \times 10^{12} \nu^{2} - 6.599 \times 10^{9} \nu + 9.926, & 0 \leq \nu \leq 10^{-3}, \\
> 9.135\,\nu^{2} - 10.863\,\nu + 1.805, & 10^{-3} < \nu \leq 1.
> \end{cases}
> \end{equation}
>
> For \textbf{ImageNet}, the corresponding piecewise functions are given as:
> \begin{equation}
> \text{Accuracy}(\nu)=\begin{cases}
> 9.966 \times 10^{14} \nu^{2} - 1.0 \times 10^{9} \nu + 75.1, & 0 \leq \nu \leq 10^{-6}, \\
> -0.332\,(\log_{10}\nu)^{2} - 5.780\,\log_{10}\nu + 50.1, & 10^{-6} < \nu \leq 1,
> \end{cases}
> \end{equation}
>
> \begin{equation}
> \text{BitOps}(\nu)=\begin{cases}
> 2.326 \times 10^{16} \nu^{2} - 2.328 \times 10^{10} \nu + 31.49, & 0 \leq \nu \leq 10^{-6}, \\
> 0.166\,(\log_{10}\nu)^{2} +0.285\,(\log_{10}\nu) + 0.146, & 10^{-6} < \nu \leq 1.
> \end{cases}
> \end{equation}
>
>
> It is noteworthy that other research studies also have such hypermarameter that need to be adjusted to get the final results. For instance, ProxylessNAS [RefY] uses an $\omega$  that controls the trade-off between accuracy and latency.  Network Slimming [RefZ] and others that prune via a penalty on batch-norm scales often try a few $\lambda$  values and pick the one that yields the desired sparsity.
> In Network Slimming [RefZ], for  CIFAR10 and SVHN datasets, the sparsity-controlling hyperparameter $\lambda$ (balancing empirical loss and sparsity) is selected via grid search over $\{10^{-3}, 10^{-4}, 10^{-5}\}$.For ImageNet,in case of VGG, it sets $\lambda = 10^{-4}$.
>
> **Please refer to the revised version for better formated response.
>
> **References:**
>
> [RefU]: Mask dino: Towards a unified transformer-based framework for object detection and segmentation, IEEE/CVF Conference on Computer Vision and Pattern Recognition, 2023
>
> [RefV]: Segmenter: Transformer for semantic segmentation, IEEE/CVF international conference on computer vision, 2021
>
> [RefW]: Semantic segmentation using Vision Transformers: A survey, Engineering Applications of Artificial Intelligence, 2023
>
> [RefS]: Timeloop: A systematic approach to dnn accelerator evaluation, IEEE international symposium on performance analysis of systems and software (ISPASS), 2019
>
> [RefT]: Accelergy: An architecture-level energy estimation methodology for accelerator designs, IEEE/ACM International Conference on Computer-Aided Design (ICCAD), 2019
>
> [RefB]: Coral AI Datasheet, Online, 2020
>
> [RefC]: Overview of Edge TPU Memory, Online, 2020
>
> [RefG]: Coral M.2 Dual Edge TPU Datasheet, Online, 2024
>
> [RefH]: CACTI 7: New tools for interconnect exploration in innovative off-chip memories, ACM Transactions on Architecture and Code Optimization (TACO), 2017
>
> [RefI]: Prospects for Memory, Keynote presentation at the Workshop on Memory-Centric High-Performance Computing (MCHPC), 2019
>
> [RefJ]: Enabling Efficient and Scalable Hybrid Memories Using Fine-Granularity DRAM Cache Management, Proceedings of ISCA, 2012
>
> [RefL]: A Flexible Precision Scaling Deep Neural Network Accelerator with Efficient Weight Combination, arXiv preprint arXiv:2502.00687, 2025
>
> [RefM]: EIE: Efficient inference engine on compressed deep neural network, ACM SIGARCH Computer Architecture News, 2016
>
> [RefN]: FINN: A Framework for Fast, Scalable Quantized Neural Network Inference on FPGAs - Documentation, Online documentation, 2023
>
> [RefO]: AMD Alveo U250 Accelerator Card, Product webpage, 2023
>
> [RefP]: Benchmarking quantized neural networks on FPGAs with FINN, arXiv preprint arXiv:2102.01341, 2021
>
> [RefQ]: ASIC vs FPGA: A Comprehensive Comparison, Wevolver, 2023
>
> [RefR]: Energy Efficient ASIC/FPGA Neural Network Accelerators, Arizona State University, 2022
>
> [RefY] ProxylessNAS: Direct Neural Architecture Search on Target Task and Hardware, ICLR 2019
>
> [RefZ]   Learning efficient convolutional networks through network slimming, international conference on computer vision, 2017

---

### Author Response · Authors · 2025-03-17
**Summary of author's response organization**

We sincerely appreciate the reviewers' insightful feedback. We are encouraged by their recognition of the novelty of our proposed solution (**g4pC, gXNN**), the comprehensive experimental results (**g4pC, gXNN, mDKj**), the clear organization and writing (**gXNN, mDKj**), the extensive related work analysis and the efficiently of the proposed solution compared to them (**mDKj**, **g4pC**).
**We have addressed the reviewers' concerns both in this response and in Appendix B of the revised paper, to the maximum practicable extent within the limitations of the submission time frame.**

---

### Decision · Action_Editor_XJHQ · 2025-03-31

**Recommendation:** Accept as is

**Comment:**

This paper presents a framework that combines architectures search and weight optimization, along with a mixed-precision search for both weighs and activations in convolution networks. The method is evaluated on CIFAR10 and ImageNet datasets, demonstrating improved performance and searching efficiency compared to previously reported methods.

The paper is well-written and has received favorable reviews from all reviewers. The reasons for acceptance are:

1). The method is reasonably novel, particularly in its extension of DARTS frameworks to support efficient mix-precision search and the introduction of complexity-aware loss functions.

2). The evaluation and ablation studies are comprehensive. The results clearly show improvement over baseline methods.

3). During the rebuttal, the authors provide detailed clarification and additional results, including the hardware performance and the extension the framework to transformer models.

In summary, this paper is technically solid and potentially useful for deploying models on edge devices. Hence, the paper is recommended for acceptance.

**Audience:**

Yes, particularly for researchers working in the area of Neural Architecture Search, quantization and deploying models on edge devices.

**Claims And Evidence:**

Yes